# Extrusion Coating of Paper with Poly(3-hydroxybutyrate-co-3-hydroxyvalerate) (PHBV)—Packaging Related Functional Properties

**Sven Sängerlaub** [1,†] , **Marleen Brüggemann** [1,2,†] , **Norbert Rodler** [1] , **Verena Jost** [1] and **Klaus Dieter Bauer** [1,*,†]

1. Fraunhofer Institute for Process Engineering and Packaging IVV, Giggenhauser Strasse 35, 85354 Freising, Germany
2. TUM School of Life Sciences Weihenstephan, Chair of Food Packaging Technology, Technical University of Munich, Weihenstephaner Steig 22, 85354 Freising, Germany
* Correspondence: klaus.dieter.bauer@ivv.fraunhofer.de
† These authors contributed equally to this work.

**Abstract:** Taking into account the current trend for environmentally friendly solutions, paper coated with a biopolymer presents an interesting field for future packaging applications. This study covers the application of the biopolymer poly(3-hydroxybutyrate-co-3-hydroxyvalerate) (PHBV) on a paper substrate via extrusion coating. The intention of this study is to analyse the effect of a plasticiser on the processability (melting point, film thickness) and the final properties (crystallinity, elongation at break) of PHBV. Up to 15 wt.% of the plasticisers triethyl citrate (TEC) and polyethylene glycol (PEG) were used as additive. The processing (including melt flow rate) as well as the structural properties (melting and crystallisation temperature, surface structure by atomic force microscopy (AFM), polarisation microscopy, scanning electron microscopy (SEM)), mechanical properties (elongation at break, tensile strength, elastic modulus, adhesion), and barrier properties (grease) of these blends and their coating behaviour (thickness on paper), were tested at different extrusion temperatures. The melting temperature ($T_m$) of PHBV was reduced by the plasticisers (from 172 °C to 164 resp. 169 °C with 15 wt.% TEC resp. PEG). The minimal achieved PHBV film thickness on paper was 30 μm owing to its low melt strength. The elastic modulus decreased with both plasticisers (from 3000 N/mm$^2$ to 1200 resp. 1600 N/mm$^2$ with 15 wt.% TEC resp. PEG). At 15 wt.% TEC, the elongation at break increased to 2.4 length-% (pure PHBV films had 0.9 length-%). The grease barrier (staining) was low owing to cracks in the PHBV layers. The extrusion temperature correlated with the grease barrier, mechanical properties, and bond strength. The bond strength was higher for films extruded with a temperature profile for constant melt flow rate at different plasticiser concentrations. The bond strength was max. 1.2 N/15 mm. Grease staining occurs because of cracks induced by the low elongation at break and high brittleness. Extrusion coating of the used specific PHBV on paper is possible. In further studies, the minimum possible PHBV film thickness needs to be reduced to be cost-effective. The flexibility needs to be increased to avoid cracks, which cause migration and staining.

**Keywords:** extrusion coating; paper coating; biopolymer; biodegradable polymer; PHBV; plasticizer

## 1. Introduction

### 1.1. Polyhydroxyalkanoate (PHA)

Polyhydroxyalkanoates (PHA) are thermoplastic polymers that are formed by bacteria [1–7]. The PHA-type polymer poly(3-hydroxybutyrate co-3-hydroxyvalerate) (PHBV) is fully biodegradable in soil and marine environments [1,8,9]. For producing PHAs, the bacteria are fed with agricultural products or by-products of food and agricultural production. The various types of PHA and derivatives differ in their polymer structure (side chains, chain length). The polymer structure influences their properties, such as mechanical, rheological, and thermal properties [2,10]. A common PHA is poly(3-hydroxybutyrate) (PHB). PHB was isolated and characterised already in 1925 [11].

PHB is highly crystalline (>70%) and thus brittle, difficult to process, and has a tendency for secondary crystallisation [12]. The melting and decomposition temperatures are close, resulting in a small thermal processing window to avoid polymer chain scission at too high temperatures [8,10]. To reduce high production costs of PHAs [5,13], which are mostly caused by substrate costs [14,15], processes are optimised in various research activities, for example, optimised fermentation and metabolic engineering [4,16].

Copolymerisation of hydroxybutyrates (HB) and hydroxyvalerates (HV) results in poly(hydroxybutyrate-co-hydroxyvalerate) (PHBV). The prolonged side chain of HV can reduce the crystallinity, the brittleness, and the melting temperature, and increase the toughness [17]. PHBV is a highly crystalline polymer [10] with a lower elastic modulus, higher elongation at break, and lower tensile strength compared with PHB [18]. Also, the processing conditions influence the final properties. An increased processing temperature during extrusion can cause lower molecular weights and, therefore, a lower elongation at break and a lower tensile strength [19].

### 1.2. Plasticisers

The processability of polymers can be improved by optimisation of processing [19], by blending with a second polymer, by additivation with plasticizers [20–27], or by using nucleation agents.

Plasticisers reduce the melting and the glass transition temperature [18] and, therefore, enable a lower processing temperature. Thereby, the brittleness can be reduced and the elongation at break can be increased. In other studies, PHBV was plasticised with triethyl citrate and polyethylene glycol [21–23,25,27]. However, extrusion of such films was rarely described, for example, by Jost [27], and coating of paper with plasticised PHBV has not been described much yet either [28,29]. The used plasticisers differ in their molecular weight (triethyl citrate 276 g mol$^{-1}$, polyethylene glycol 950–1050 g mol$^{-1}$). PHBV and polyethylene glycol are miscible in all mass ratios, except 50/50 [25]. Higher concentrations of plasticiser increase the water vapour permeability and the velocity of biodegradation [23].

### 1.3. Motivation and Intention of this Study

One application scenario for papers coated with PHBV is service paper and packaging for "to go food". Such materials should be biodegradable for the case of a release in the environment with no proper disposal or recycling. The biodegradability of PHBV can avoid an accumulation of microplastic in the environment.

Extrusion coating on paper with plasticised PHBV has rarely been examined thus far. Published results about such materials, without plasticiser, are from the year 2000 [28,29]. The aim of this study was to present new results for the extrusion coating of paper with PHBV as a possible substitute for other, non-biodegradable, and non-biobased polymers, and to optimise processing methods and formulations.

## 2. Materials and Methods

*2.1. Materials*

### 2.1.1. PHBV

The used PHBV had no additives (trade name PHI002; injection moulding grade; NaturePlast, France). The polymer has the following properties (according to the material data sheet): density 1.23 g/cm$^3$, elastic modulus 4200 MPa, tensile elongation at break 3 length-%, melt flow rate at 190 °C, and 2.16 kg 5–10 g/10 min. The hydroxyvalerate concentration is 5–8% according to the supplier.

### 2.1.2. Plasticisers

Two plasticisers were taken: triethyl citrate (TEC, trade name 800251; $M_w$ = 276 g/mol, $T_m$ = −55 °C, decomposition temperature > 200 °C, density 1.135 g/cm$^3$) and polyethylene glycol (PEG, trade name 03395; Sigma-Aldrich Chemie GmbH, Munich, Germany; $M_w$ = 950–1050 g/mol, $T_m$ = 33–40 °C, 1.2 g/cm$^3$).

The specific polyethylene glycol was taken for good processability, considering the results of Jost [27]. Both TEC and PEG up to a molecular weight of 1500 g/mol are biodegradable [30,31], which was a prerequisite for their use in this study.

### 2.1.3. Paper Substrate

As substrate, a 50 μm thick paper (Niklakett Spezial TD-M, Brigl & Bergmeister GmbH, Niklasdorf, Austria) was used. The PHBV was coated on the functional side of the paper. It was stored at 50% relative humidity at 23 °C.

*2.2. Methods*

### 2.2.1. Compounding

Before compounding, the PHBV was dried (type Heliomat 2T; HELIOS GmbH, Rosenheim, Germany) with an air volume flow rate of 15 m$^3$/h for 24 h at 60 °C.

PHBV was combined with TEC and PEG by a compounding step. The used compounder was a twin screw-extruder (type ZK 25 T × 18/24; Dr. COLLIN Lab & Pilot Solutions GmbH, Maitenbeth, Germany). TEC and PEG were added into the melted PHBV in the compounder using a syringe pump (type DX100; Teledyne Isco, NE, USA). The delivery rate of a pump for TEC was 30.7, 79.5, 167, and 265.5 mL/h for a TEC concentration of 2, 5, 10, and 15 wt.%, respectively. For PEG, the delivery rate of the pump was 30, 75, 158.3, and 250 mL/h for a PEG concentration of 2, 5, 10, and 15 wt.%, respectively. The delivery rate of PEG was lower because of the higher density of the used PEG. The PEG was heated and melted at circa 40 °C. The temperature profile of the compounder during compounding was as follows: 18, 70, 160, 160, 150, 150, 155 °C. The rotational frequency was 100 rotations/min and the melt pressure was ~40 bar.

The polymer strand was cooled in water and cut to pellets with a granulator (type CSG 171 T; Dr. COLLIN Lab & Pilot Solutions GmbH, Maitenbeth, Germany) with the following parameters: take-off speed 20 m/min; cutting length 2.5 mm; granule diameter < 4 mm, length of the water bath 150 cm.

### 2.2.2. Film Extrusion and Paper Coating

Before extrusion, the PHBV blend was dried (in cotton pouches) as described in the compounding section.

Between every trial, PHVB polymer was taken for flushing. For the extrusion, a single screw extruder (type E 25 P × 25 D, 24; Dr. COLLIN Lab & Pilot Solutions GmbH, 83558 Maitenbeth, Germany; screw diameter 25 mm, screw length 25 D) with a rotational frequency of 30 rotations/min was used. Two temperature profiles were applied: one with a constant temperature profile and one

with a constant melt flow rate of the blend of 20.65 cm$^3$/10 min (see Section 3.3). The temperature profiles are shown in Table 1.

**Table 1.** Applied temperature profiles during extrusion for paper coating. TEC, triethyl citrate; PEG, polyethylene glycol; MFR, melt flow rate.

| Plasticiser | Concentration/wt.% | Temperature Profile/°C | |
|---|---|---|---|
| no plasticiser | 0 | constant temperature; constant MFR of 20.65 cm$^3$/10 min | 40-155-185-185-185-185 |
| TEC | 2, 5, 10, 15 | constant temperature | 40-155-185-185-185-185 |
| TEC | 2 | constant MFR of 20.65 cm$^3$/10 min | 40-155-180-180-180-180 |
| TEC | 5 | constant MFR of 20.65 cm$^3$/10 min | 40-155-179-179-179-179 |
| TEC | 10 | constant MFR of 20.65 cm$^3$/10 min | 40-155-177-177-177-177 |
| TEC | 15 | constant MFR of 20.65 cm$^3$/10 min | 40-155-176-176-176-176 |
| PEG | 2, 5, 10, 15 | constant temperature | 40-155-185-185-185-185 |
| PEG | 2 | constant MFR of 20.65 cm$^3$/10 min | 40-155-182-182-182-182 |
| PEG | 5 | constant MFR of 20.65 cm$^3$/10 min | 40-155-180-180-180-180 |
| PEG | 10 | constant MFR of 20.65 cm$^3$/10 min | 40-155-177-177-177-177 |
| PEG | 15 | constant MFR of 20.65 cm$^3$/10 min | 40-155-176-176-176-176 |

The extruded film was coated on paper and the laminate was pressed between lamination rolls (temperature 25 °C to 40 °C) of the extruder setup with a pressure of 50–60 bar. The PHBV side was covered with siliconized paper to avoid sticking. The take-off speed was varied to 1.6, 2, 2.4, 2.8, 3.2, 3.6, and 4.0 m/min. The take-off speed (coating speed) was varied because a higher coating velocity causes a lower coating thickness.

At the samples with 10 wt.% TEC and 15% wt.% PEG, a coating velocity of 4 m/min did not result in successful coating of paper, because of a too low melt film stability and tear of the melt film and instable extrusion.

After extrusion, the films were stored at 23 °C and 50% relative humidity. The first measurements were done 24 h after extrusion.

### 2.2.3. Film Thickness

The film thickness was measured with a film thickness testing device (Precision Thickness Gauge FT3, Hanatek Instruments, Hastings, UK). For the mechanical tests, 10 specimens (stripes) were cut in machine direction and each was measured five times. The mean values and standard deviations were calculated. For the other measurements, where the thickness is required, two specimens (stripes) were cut and measured 10 times each.

### 2.2.4. Differential Scanning Calorimetry (DSC)

With DSC (type DSC 3 + STARe System, Mettler-Toledo GmbH, Gießen, Germany), the melting ($T_m$), the glass transition ($T_g$), and the crystallisation ($T_c$) temperature was measured, according to DIN EN ISO 11357-1. The following parameters were taken: sample weight 5 to 15 mg; in gaseous nitrogen; heating rate 10 °C/min; temperature from −10 to 200 °C. The samples were analysed at day one, day 14, and day 28 after production as single measurements. The samples were exposed to two DSC cycles. The second cycle was used for evaluation. The second cycle was taken in order to analyse samples exposed to a known thermal treatment and to avoid artefacts such as those from thermal parameter variations during processing and such as cooling rate and temperature variations.

For the estimation of the heat of fusion of a PHB-crystal, a value of 146 J/g was used [32].

PEG was also analysed at the first heating cycle from 10 to 100 °C, cooled to −10 °C, and then heated to 200 °C with 10 °C/min.

### 2.2.5. Thermogravimetric Analysis (TGA)

TGA was externally measured (Hochschule Bremerhaven, Bremerhaven, Germany) at a maximum temperature of 1000 °C and a heating rate of 50 °C/min to analyse the decomposition behaviour. The samples were flushed with nitrogen during the test.

### 2.2.6. Melt Flow Rate (MFR)

The melt flow rate was measured with a melt index tester (type MeltFloW; Emmeram Karg Industrietechnik, Planegg, Germany) with 5 g sample per test. One sample was measured 10 times. The measuring weight was 2.16 kg and the preheating time was 300 s. The testing temperature was varied to study the dependence of the MFR on the melt temperature.

### 2.2.7. Mechanical Properties

The mechanical properties were tested with a tensile testing machine (type Z005, Allround Line; Zwick GmbH & Co. KG, Ulm, Germany). The tensile tests (elastic modulus, elongation at break, tensile strength) were performed at 23 °C and 50% relative humidity according to DIN EN ISO 527. The samples were at least 48 h conditioned before the test [33]. Before testing the PHBV film, it was separated from the paper substrate. The samples were cut to a length of 200 mm and a width of 15 mm. For every test ten specimens, with and across machine direction, were prepared. The thickness of each specimen was measured five times. The following parameters were applied: clamp distance 50 mm, clamp pressure two bars, initial load 0.2 N, and a testing velocity 50 mm/min. Outlier values were identified by a Grubbs test and taken out. In the Tables S1–S3 the values for the elastic modulus, the elongation at break and the tensile strength are presented.

### 2.2.8. PHBV Layer Bond Strength on Paper

The bond strength was tested with a tensile testing device (DOLI GmbH Industrie Elektronik München, Germany). The samples were tested 10 times in machine direction (MD) and 10 times in cross direction (CD) of the films. The sample stripes had a width of 15 mm. The samples were conditioned before testing for 48 h at 23 °C and 50% relative humidity. The end of the sample had an angle of 90° to the clamps. The following parameters were used: clamp distance 50 mm. Outliers were taken out after identification with the Grubbs test.

### 2.2.9. Grease Barrier (Staining)

The grease barrier was measured according to a method developed by Fraunhofer IVV (Freising, Germany). For every sample, five measurements were taken. At five spots of the film, an area of 25 cm$^2$ was marked. The spot was coated with peanut oil with Sudan Red (EC-No.: 2,216,477) and then covered by a fleece of the same size. The preparation was stored for 48 h at 23 °C and 50% relative humidity. After this time, the fleece was removed and the surface was cleaned. When the oil penetrated the sample, spots were visible on the counter side. Pictures were taken and the stained area was evaluated by a drawing program, resulting in the percentage of stained area. Outlier values were identified by Grubbs test and taken out. In the Table S4 the values for the fat penetration (staining) are presented.

### 2.2.10. Surface Roughness

The surface roughness was tested with a surface roughness testing device (type Hommel-Etamic W55 Version 1.0.9, sensing type TK100; Hommelwerke, Schweitenkirchen, Germany). Five measurements were performed in machine direction and five in cross direction. From the results, the mean values and the standard deviations were calculated. The following parameters were applied: measuring range 80 µm, measuring distance 4.8 mm, measuring velocity 0.5 mm/s, cut off 0.8 mm.

### 2.2.11. Atomic Force Microscopy (AFM)

With AFM, the surface roughness of samples can be measured. With AFM (type alpha 500; WITec GmbH, Ulm, Germany), samples of a size of 50 µm × 50 µm were analysed. The centre of the samples was analysed with a higher resolution with an area of 20 µm × 20 µm. The following parameters were applied: amplitude 0.08 to 0.15, setpoint 0.8 V, P-Gain 8%, I Gain 4%, 50 µm × 50 µm with 2 s/line and 2 s minimum time for retrace, 20 µm × 20 µm with 1 s/line and 1 s minimum time for retrace.

### 2.2.12. Polarisation Microscopy

The samples had a thickness of 5 µm (microtome cut). The magnification was 200 and 500. For the analysis, a threshold value for brightness of 125 was set as well as a constant area for the samples (1890 × 290 µm). For the evaluation of structure size (crystallite sizes), a histogram with 20 classes was set up. Pictures were taken with a camera (Diaplan with universal condensator, camera type DFC 295; Leica Microsystems, Wetzlar, Germany).

### 2.2.13. Scanning Electron Microscopy (SEM)

The film samples were sputtered with a 10 nm thick gold layer to avoid electrical charging. The pictures were taken with an SEM (type JSM-7200F; JEOL Ltd., Tokyo, Japan). A secondary electron detector was used.

### 2.2.14. Raman Spectroscopy

With a Raman microscope (type alpha 500; WITec GmbH, Ulm, Germany; CCD camera DU970N-BV, EMCCD; Spectrometer UGTS 300, WITec GmbH, Ulm, Germany; software WITec Control and WITec Project 4.1), mapping and an in depth scan were done. Using cluster analysis, the surface was analysed. With crop-function, the area above the film was removed for an underground analysis, which is only valid for the spectrum of the film.

## 3. Results and Discussion

### 3.1. Melting and Crystallisation Temperature

In Figures 1 and 2, the melting and the crystallisation temperatures of the produced samples are shown. The measurements are from day 1, 14, and 28 after production. The points in the figures are medium values from the measurements. We expected a measurable time dependent change of these values, but differences were not significant. Therefore, we decided to take the medium values from these three measurement days. At these single days, single measurements were done. For the melting temperatures, the results of other studies [21,22,27] are also inserted for comparison.

With increasing plasticiser concentration, the melting and the crystallisation temperatures decrease. The reason is a plasticising effect and a higher polymer chain mobility caused by the plasticisers [18]. Other studies found comparable trends [21,27]. The melting point also decreased with increasing plasticiser concentration, but with a bigger slope (Choi [21]) compared with this study. The differences of Choi [21] and Rodrigues [22] compared with this and Jost's study can be attributed to the lower molecular weight of PEG (300 g/mol) [22]. Additionally, both samples (from Choi and Rodrigues) were cast films from chloroform, which results in a different polymer structure.

Because of the reduction of the $T_m$, the extrusion process can be performed at lower temperatures, resulting in a lower temperature impact on the polymer. Rodrigues explained the reduced crystallisation temperature by hindered nucleation as a result of the reduced crystallisation rate based on the impact of the plasticiser. This causes the formation of smaller spherulites and explains the higher flexibility of the blends compared with the pure polymer [22].

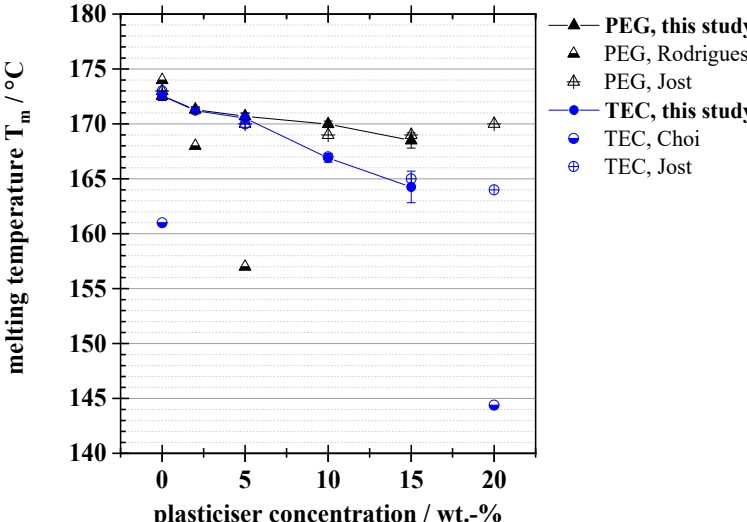

**Figure 1.** Melting temperatures of poly(3-hydroxybutyrate-co-3-hydroxyvalerate) (PHBV) and PHBV with different concentrations of plasticizers. PEG: polyethylene glycol, TEC: triethyl citrate.

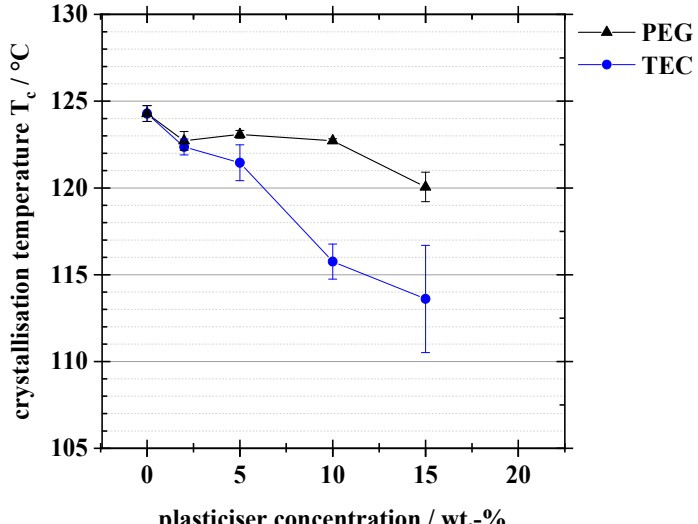

**Figure 2.** Crystallisation temperature of PHBV and PHBV with different concentrations of plasticizers. PEG: polyethylene glycol, TEC: triethyl citrate.

The crystallinity of pure PHBV was similar to the results of Jost [27], with lower values compared with another study with a value of 67% [34]. The difference can be explained by the different PHBV grade used.

As expected, the crystallinity is reduced by the addition of plasticisers. On the basis of PHBV's strong tendency for a secondary crystallisation, an increase of crystallinity as a function of time was expected, affecting the mechanical properties [35]. However, the crystallinity stayed constant or decreased during storage time (see Figures 3 and 4, single measurements). Our assumption is that samples absorbed water, acting as a plasticizer and affecting the structure. Samples were stored at 50% relative humidity and 23 °C. However, this hypothesis was not further verified. Furthermore, we cannot explain the observed differences in post-processing crystallization behaviour.

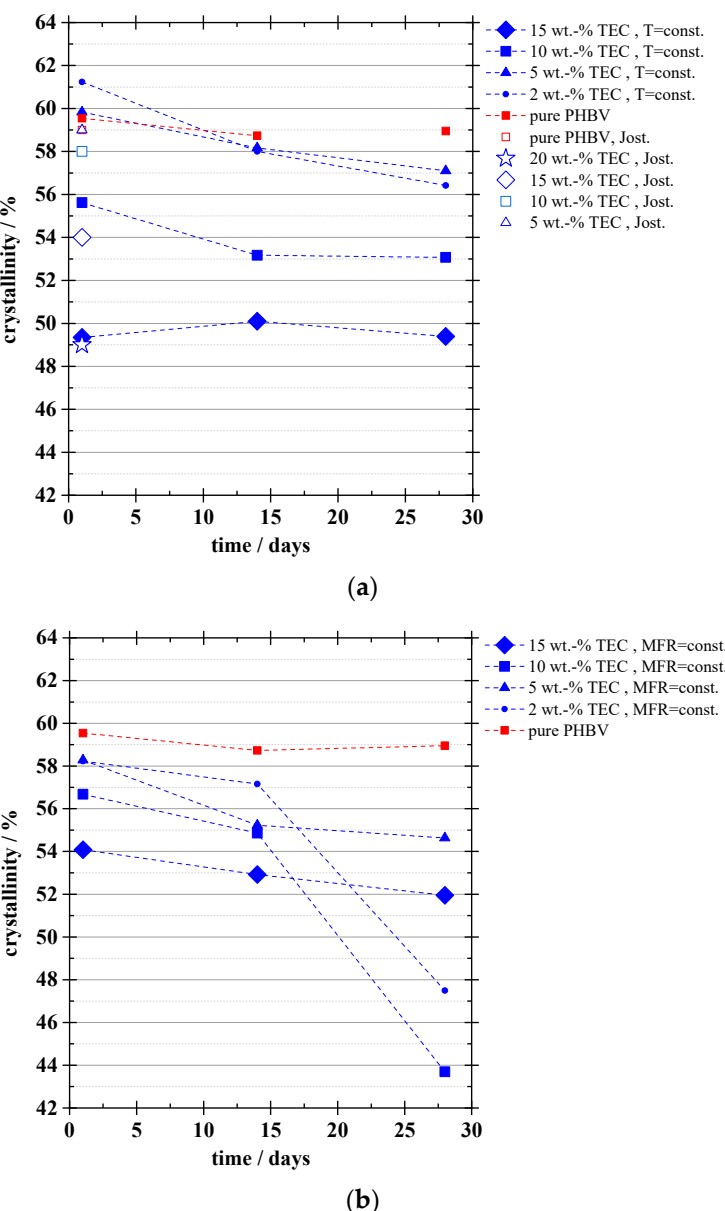

**Figure 3.** Crystallinity as function of time of PHBV layers blended with TEC: (**a**) temperature (T) during extrusion constant; (**b**) melt flow rate (MFR) during extrusion constant.

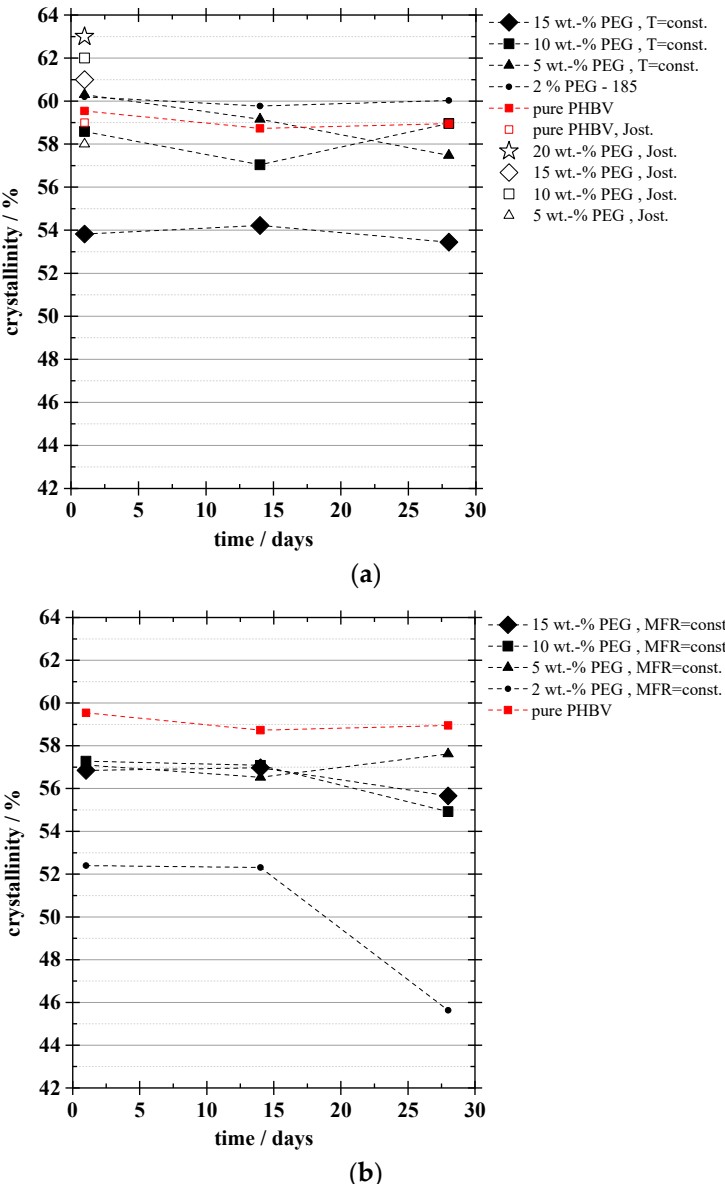

**Figure 4.** Crystallinity as function of time of PHBV layers blended with PEG: (**a**) temperature (T) during extrusion constant; (**b**) melt flow rate (MFR) during extrusion constant.

By DSC analysis, it was shown that the melting peak at the second heating becomes larger and narrower, because the speed of the thermal ramp has allowed a good crystallization (Figure 5). During DSC analysis, samples with 15 wt.% PEG had a peak at circa 40 °C only in the first cycle (Figure 5). This peak represents the $T_m$ of PEG (33–40 °C). A possible reason for why the PEG peak at 40 °C diminished can be the dissolution and a more consistent distribution of PEG in PHBV during the first heating cycle. PEG could, in such a case, not crystallize at the cooling rate of 10 °C/min. This observation is an indication of a time-dependent molecular separation of PEG and PHBV after extrusion. The literature describes a decomposition of PEG 1500 at temperatures above 175 °C (with a maximum between 200 and 250 °C) [36]. However, in the case of a decomposition of PEG, a higher $T_m$ of PHBV with 15 wt.% PEG would be expected. Furthermore, the shape of the PHBV fusion peak should have changed and become higher and narrower. Therefore, decomposition of PEG is not an explanation, or only a part of the PEG decomposed. Furthermore, the TGA results do not back the explanation for a decomposition of PEG (see Section 3.2.).

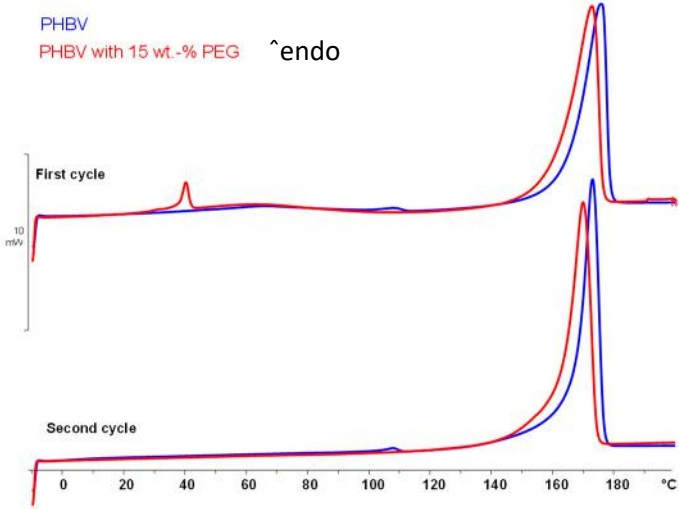

**Figure 5.** First and second heating cycle of a differential scanning calorimetry (DSC) measurement.

*3.2. Decomposition*

Decomposition is determined with TGA by a weight loss. In this study, PHBV and blends were already exposed to thermal stress by compounding. TGA results, analysed under a nitrogen atmosphere (Table 2), cannot be directly transferred to PHBV extrusion with available oxygen. PHBV decomposed at 300 °C, much higher than the applied extrusion temperature of 185 °C. Chiellini [37] observed an onset temperature, that is, decomposition, at 270 °C for PHBV with unknown HV concentration. The decomposition temperature decreases by the addition of plasticisers, PHBV with PEG decomposes at 290 °C. This is in opposition to the literature reporting no influence of PEG ($M_w$ 300 g/mol) on thermal stability of a PHB–PEG blend [22]. Here, however, the PEG had a molecular weight of 300 g/mol and PHB instead of PHBV was analysed.

**Table 2.** Decomposition temperatures of pure and plasticised poly(3-hydroxybutyrate-co-3-hydroxyvalerate) (PHBV) measured with thermogravimetric analysis (TGA). TEC: triethyl citrate, PEG: polyethylene glycol.

| Plasticiser | Concentration/wt.% | Onset Temperature/°C |
|:---:|:---:|:---:|
| no plasticiser | 0 | 300 |
| TEC | 2 | 257 |
| TEC | 5 | 261 |
| TEC | 10 | 264 |
| TEC | 15 | 268 |
| PEG | 2 | 209 |
| PEG | 5 | 171 |
| PEG | 10 | n.a. |
| PEG | 15 | 292 |

The concentration of HV also affects the decomposition behaviour, reducing the decomposition temperature as the concentration of HV increases [38]. On the other hand, the starting temperature for decomposition of the PHBV with a comparable HV content (5%), but from different suppliers varies between 187 °C and 238 °C.

*3.3. Melt Flow Rate*

Plasticising PHBV reduces the $T_m$ as well as the viscosity. To achieve similar processing of pure and plasticised PHBV, they were extruded either at constant melt flow rate and different temperatures or at a constant temperature (see Table 1). As constant melt flow rate, the melt flow rate of pure PHBV at 185 °C was taken, with a value of 20.65 ± 0.88 cm$^3$/10 min. To identify the corresponding temperature, the melt

flow rate was measured as a function of the temperature (Figure 6). For identification of the extrusion temperature, a linear correlation is applied (Table 3). Increasing the plasticiser concentration increases the MFR of plasticised PHBV. The increase is stronger with PEG compared with TEC. Therefore, PEG-plasticised PHBV is less viscous at higher temperatures, affecting film extrusion. A lower viscosity can also be induced by a thermal degradation due to a long residence time in the extruder [19,39].

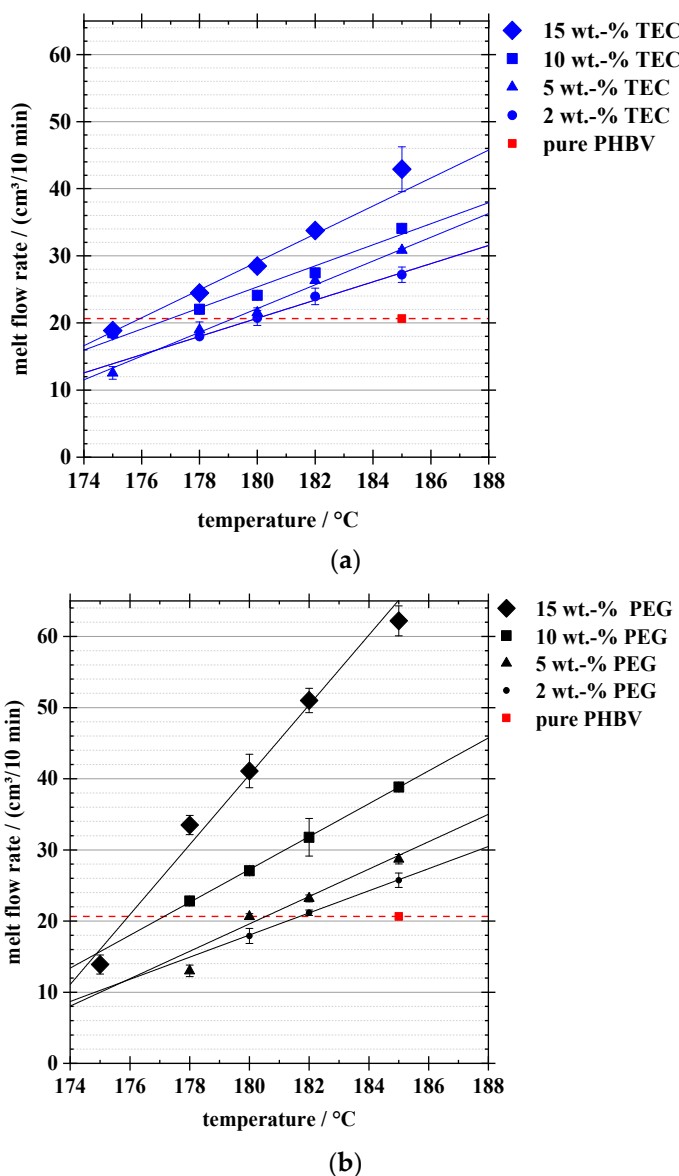

**Figure 6.** Melt flow rate of PHBV: (**a**) blended with PEG; (**b**) blended with TEC.

**Table 3.** Temperatures for constant melt flow rate. TEC: triethyl citrate, PEG: polyethylene glycol.

| Plasticiser | Concentration/wt.% | Temperature/°C |
|---|---|---|
| no plasticiser | 0 | 185 |
| TEC | 2 | 180 |
| TEC | 5 | 179 |
| TEC | 10 | 177 |
| TEC | 15 | 176 |
| PEG | 2 | 182 |
| PEG | 5 | 180 |
| PEG | 10 | 177 |
| PEG | 15 | 176 |

*3.4. Mechanical Properties: Tensile Strength, Elongation at Break, Elastic Modulus*

For this analysis, the PHBV layers were peeled off the paper substrate. With increasing plasticiser concentration, the tensile strength decreases (Figure 7). After reaching the maximal value, the samples broke, which is a characteristic of brittle materials. The high error bars are an indication of inhomogeneity of the samples. The samples with TEC had no significant difference between the two processing conditions over all plasticiser concentrations. This is not in accordance with the literature [21,27]. The samples with PEG have a higher tensile strength with a constant melt flow rate (equals a lower extrusion temperature) compared with the samples with constant extrusion temperature. This might be because of the higher thermal degradation at processing with constant extrusion temperature. The reduction of tensile strength with an increasing PEG concentration is in accordance with the literature [23,27]. The tensile strength of pure PHBV (around 31 N/mm$^2$) is lower compared with the literature (36 N/mm$^2$ [34], 31.3 N/mm$^2$ [27], 23.5 to 28 N/mm$^2$ [27]). Choi determined a value of 15 N/mm$^2$ for PHBV blends with 20 wt.% TEC, for pure PHBV of 45 N/mm$^2$. Differences can be explained by the different PHBVs used. Other polymers that can be used for coating of paper, such as high-density polyethylene (PE-HD), have a higher tensile strength of 30 N/mm$^2$ to 40 N/mm$^2$ [40].

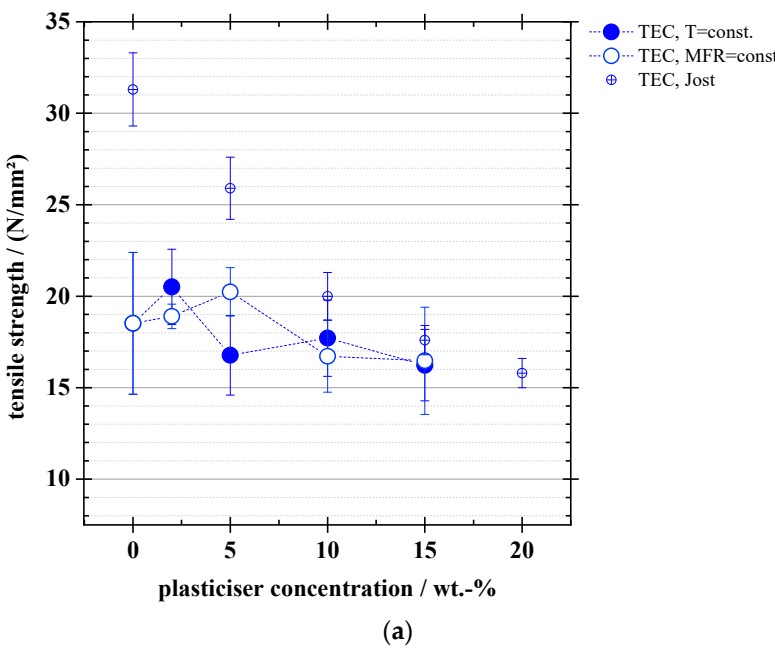

**(a)**

**Figure 7.** *Cont.*

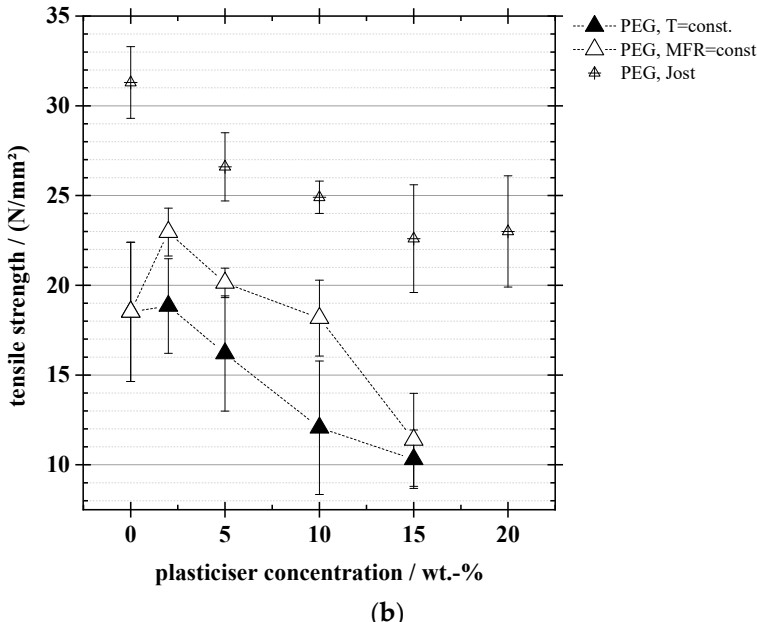

(**b**)

**Figure 7.** Tensile strength as function plasticizer concentration: (**a**) of triethyl citrate (TEC) concentration; (**b**) of polyethylene glycol (PEG) concentration. Comparison with results of Jost. MFR: melt flow rate during extrusion; T: temperature during extrusion.

The elongation at break is an important property of PHBV coated paper because it influences the formation of cracks during bending. The highly crystalline PHBV has a very low elongation at break. The elongation at break of the pure material is slightly higher or lower than the values reported by the literature of 0.8 length-% [27] and 1 length-% [34], or in a similar range of 0.8 to 1.2 length-% [27]. With increasing TEC concentration, the elongation at break of the films increased (Figure 8). Also, with increasing PEG concentration, at constant MFR, the elongation increases (Figure 8). PEG does not seem to significantly affect the elongation at break considering the large value of the error bars there.

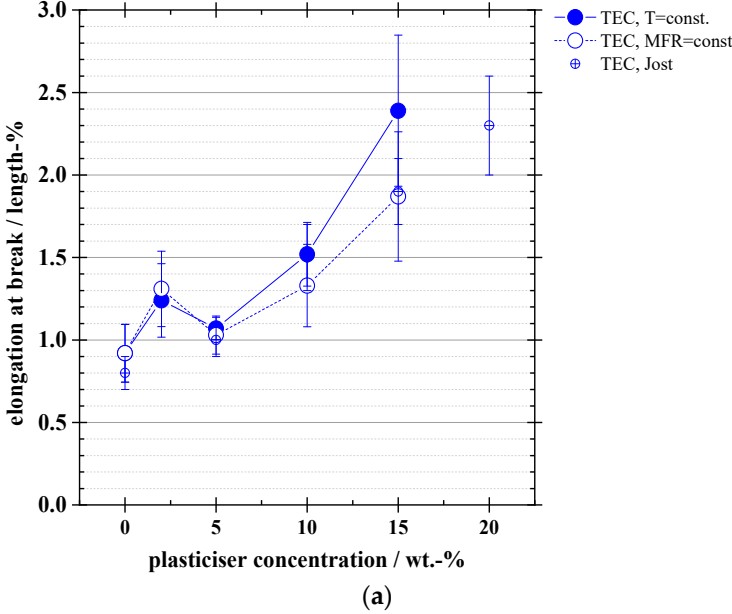

(**a**)

**Figure 8.** *Cont.*

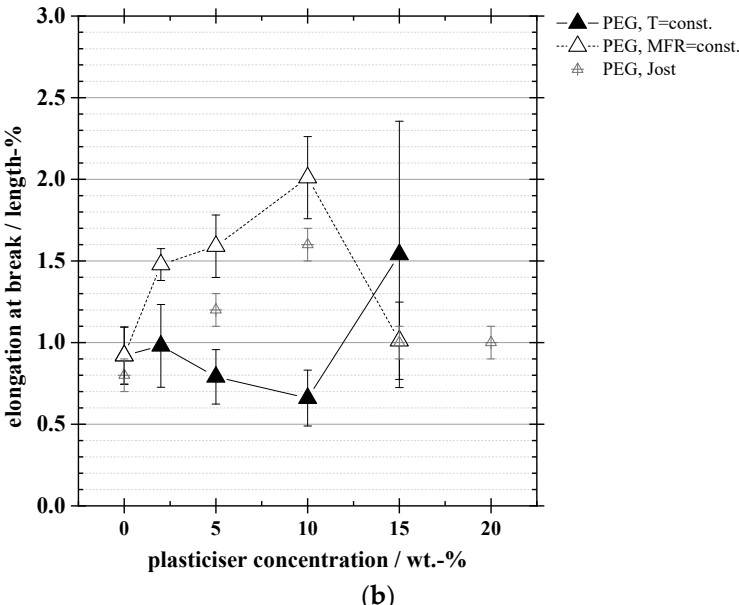

(**b**)

**Figure 8.** Elongation at break as function of plasticizer concentration: (**a**) of TEC; (**b**) of PET. Comparison with results of Jost. MFR: melt flow rate during extrusion; T: temperature during extrusion.

Parra also found a higher elongation at break. For blends from PHA and 2 to 10 wt.% of PEG, elongation was 25 length-%, compared with 9 length-% for pure PHA [23]. However, Parra produced the tested samples in another way by dissolving in chloroform and casting.

At a low plasticiser concentration of 2 wt.%, the elastic modulus slightly increased, possibly because of better film forming properties (Figure 9). A further increase of both plasticiser concentration leads to a significant decrease of the elastic modulus, as expected. The elastic modulus of pure PHBV material was lower than that reported in literature of 4400 N/mm$^2$ [34] and 3760 N/mm$^2$ [27], or in the same range of 2830 to 2950 N/mm$^2$ [27].

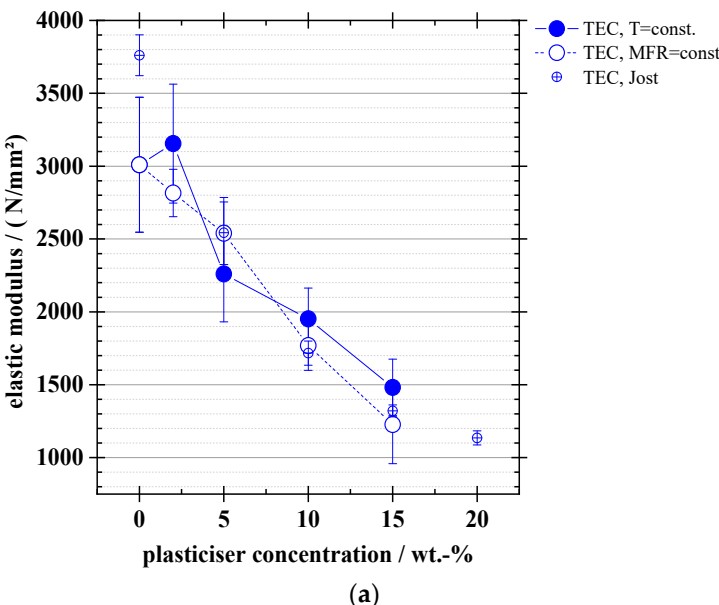

(**a**)

**Figure 9.** *Cont.*

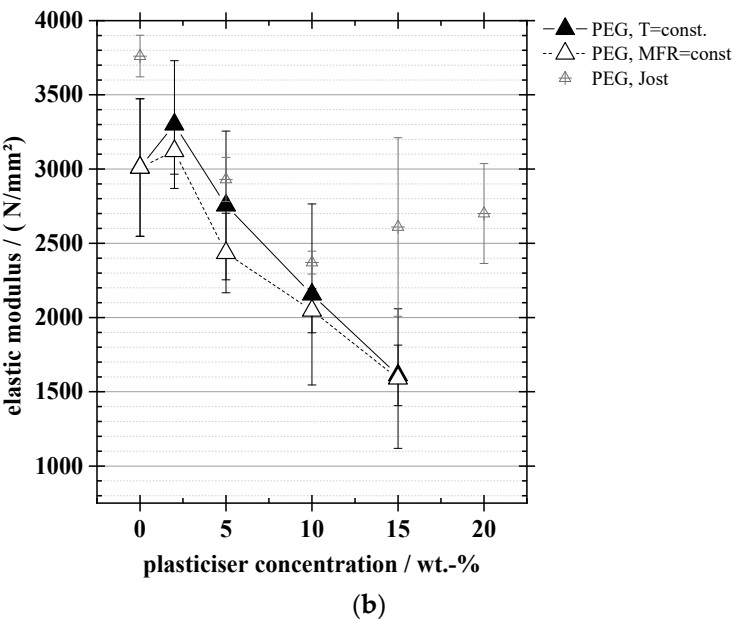

(**b**)

**Figure 9.** Elastic modulus as function of plasticizer concentration: (**a**) of triethyl citrate (TEC) concentration; (**b**) of polyethylene glycol (PEG) concentration. Comparison with results of Jost. MFR: melt flow rate during extrusion; T: temperature during extrusion.

Figure 10 shows the elongations at break as a function of elastic moduli. The results give evidence that a higher plasticiser content leads to films with increased elongation and reduced elastic modulus. Interestingly, the different processing does not significantly affect the mechanical properties of TEC-plasticised films, but does for PEG-plasticised films.

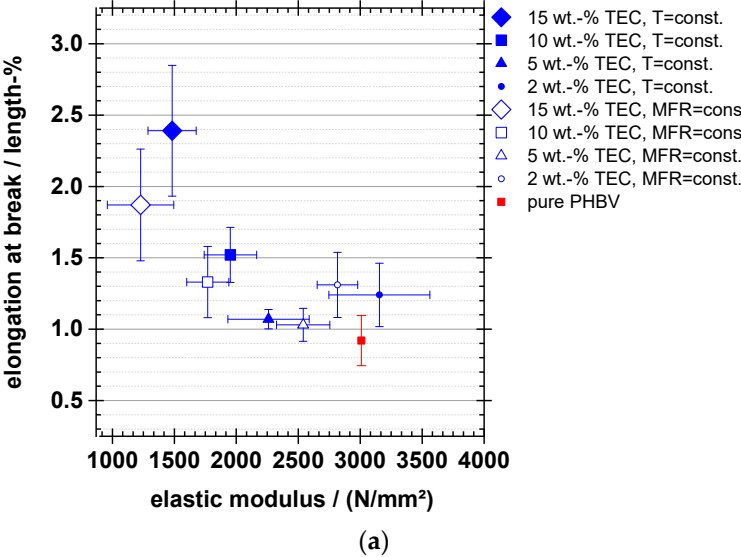

(**a**)

**Figure 10.** *Cont.*

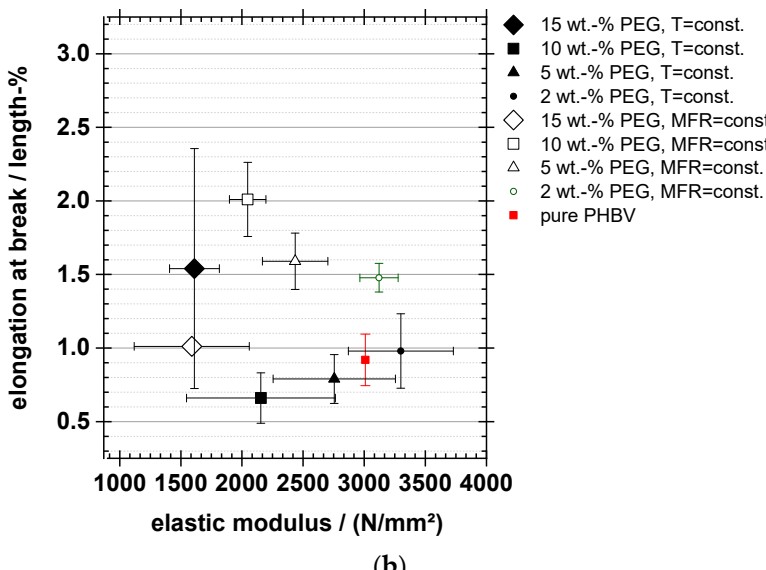

(**b**)

**Figure 10.** Elongation at break versus elastic modulus as function of plasticizer concentration: (**a**) of TEC; (**b**) of PEG. Comparison with results of Jost. MFR: melt flow rate during extrusion; T: temperature during extrusion.

### 3.5. Bond Strength of Pure and Plasticised PHBV on Paper

Bond strength is a measure to evaluate the release of PHBV off the paper substrate. According to Kuusipalo, the coating thickness influences the bond strength [28]. Increasing the TEC concentration resulted in a trend for a reduced bond strength (Figure 11). The PHBV films with TEC extruded with constant melt flow rate yielded a higher bond strength. A possible explanation is the more stable extrusion processing of these samples. At the PHBV films with 15 wt.% TEC, not all samples could be measured because of instable films during extrusion coating. At the samples with PEG, with higher plasticiser concentration, the bond strength reduced in most samples (Figure 11). At bond strengths of 0.8 to 1.0 N/15 mm, fibres teared off from the paper surface. At a bond strength of >1 N/15 mm, the paper teared and the PHBV layer did not release from the paper substrate. Therefore, at bond strengths of >1 N/15 mm, error bars with a higher value were overserved because of inhomogeneous tear off of PHBV layers from the paper. Kuusipalo defined a bond strength sufficient when a polymer layer on paper tears out fibres from the paper surface during tearing of the film [28]. However, Kuusipalo did not observed fibre tear in opposition to this work [28]. Therefore, bond strength values of ≥1 achieved in this study can be considered as sufficient for coated paper. Kuusipalo observed a higher bond strength with a higher layer thickness on paper [28]. The results of our study do not confirm this observation. In our study, the bond strength and layer thickness do not correspond.

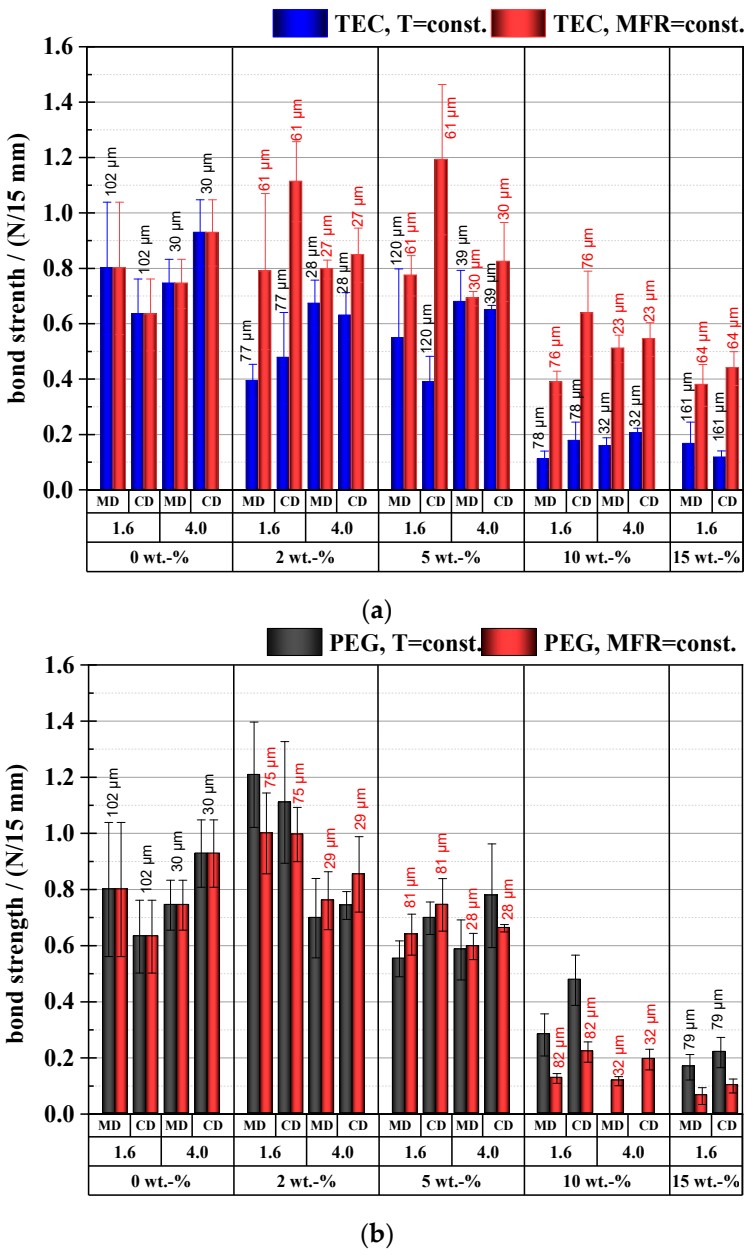

**Figure 11.** Bond strength of PHBV layers blended with plasticizer on paper: (**a**) of triethyl citrate (TEC); (**b**) of polyethylene glycol (PEG). Plasticizer concentration 0, 2, 5, 10, 15 wt.%; coating velocity: 1.6 and 4 m/min. MD: machine direction; CD: cross direction; MFR: melt flow rate during extrusion; T: temperature during extrusion, numbers above the bars: thickness in μm.

Bond strength reduces with a higher plasticiser concentration owing to a reduced adhesion [41]. The accumulation of plasticiser in the boundary layer (between PHBV and substrate) is analysed by Raman spectroscopy with the PEG samples. By the results no significant difference between pure and plasticised PHBV (Figure 12) and of plasticised PHBV analysed at different depths (Figure 13) is shown. According to these results, we cannot support the thesis of an accumulated plasticiser in the boundary layer. However, our measurement allows no determination of a wetting of the PHBV surface by PEG in the nanometre scale, which could already cause a weak boundary layer. In the Figure S1 additional RAMAN spectra of PHBV are presented.

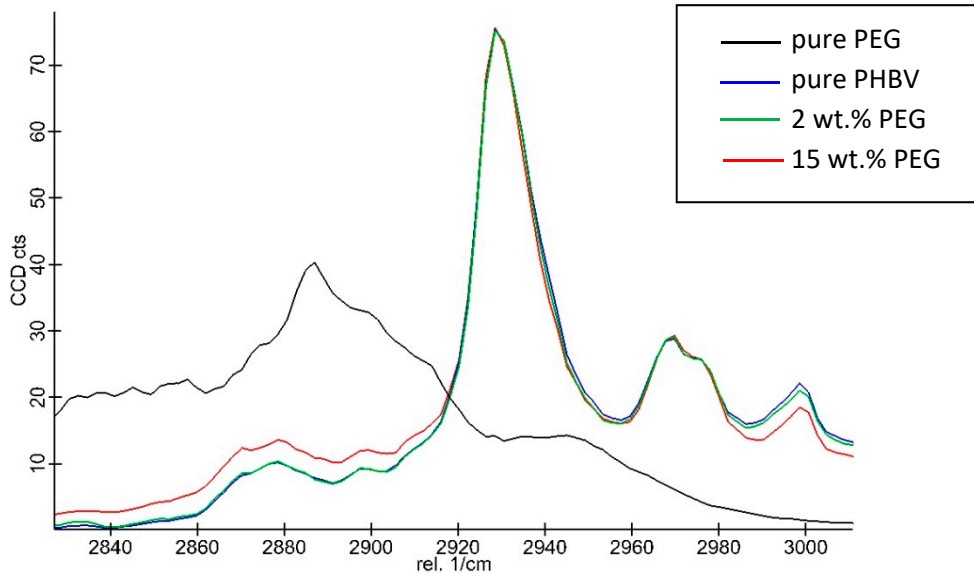

**Figure 12.** Raman spectra of samples.

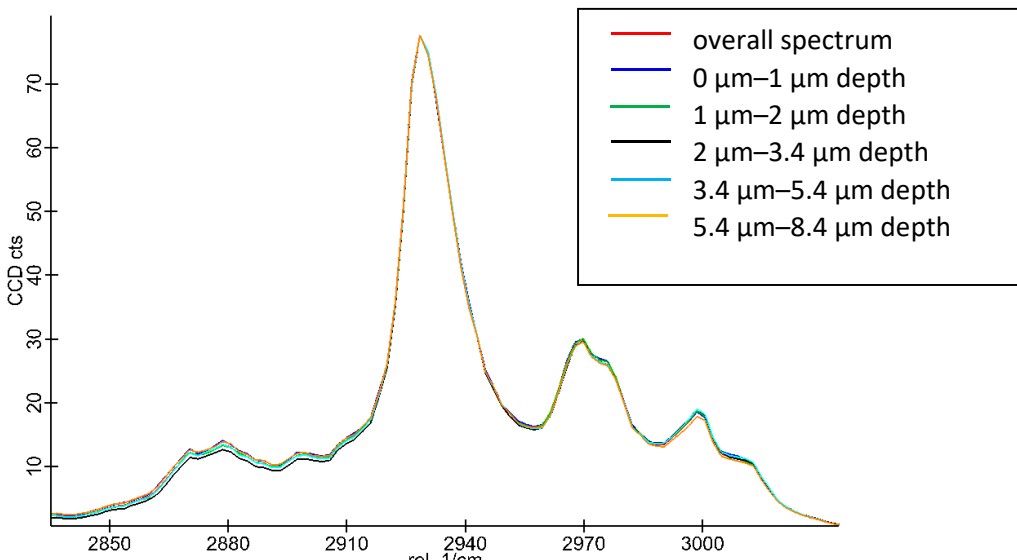

**Figure 13.** Raman spectra of PHBV with 15 wt.% PEG at different sample depths.

### 3.6. Grease Barrier (Staining)

Grease barrier is an important property to avoid staining, and is relevant, for example, for service paper. The values are the percentage of area that was stained by the coloured peanut oil. The extrusion velocity was constant for all samples, but the thicknesses varied. Therefore, the staining is depicted as a function of the PHBV-layer thickness. Our results indicate that the staining correlates with the extrusion temperature (Figures 14 and 15). At the higher extrusion temperature (T = const.), a greater area was stained. At all test rows, the samples with 2 wt.% plasticiser resulted in the lowest staining. This means a low plasticisation is sufficient to reduce defects, but remains the grease barrier property. However, the high error bars have to be mentioned, which might be a result of defects in the film. TEC as plasticiser resulted in general in lower staining. A higher film thickness resulted in less staining at all samples. As will be later shown by SEM measurements, the staining was caused by cracks in the layers.

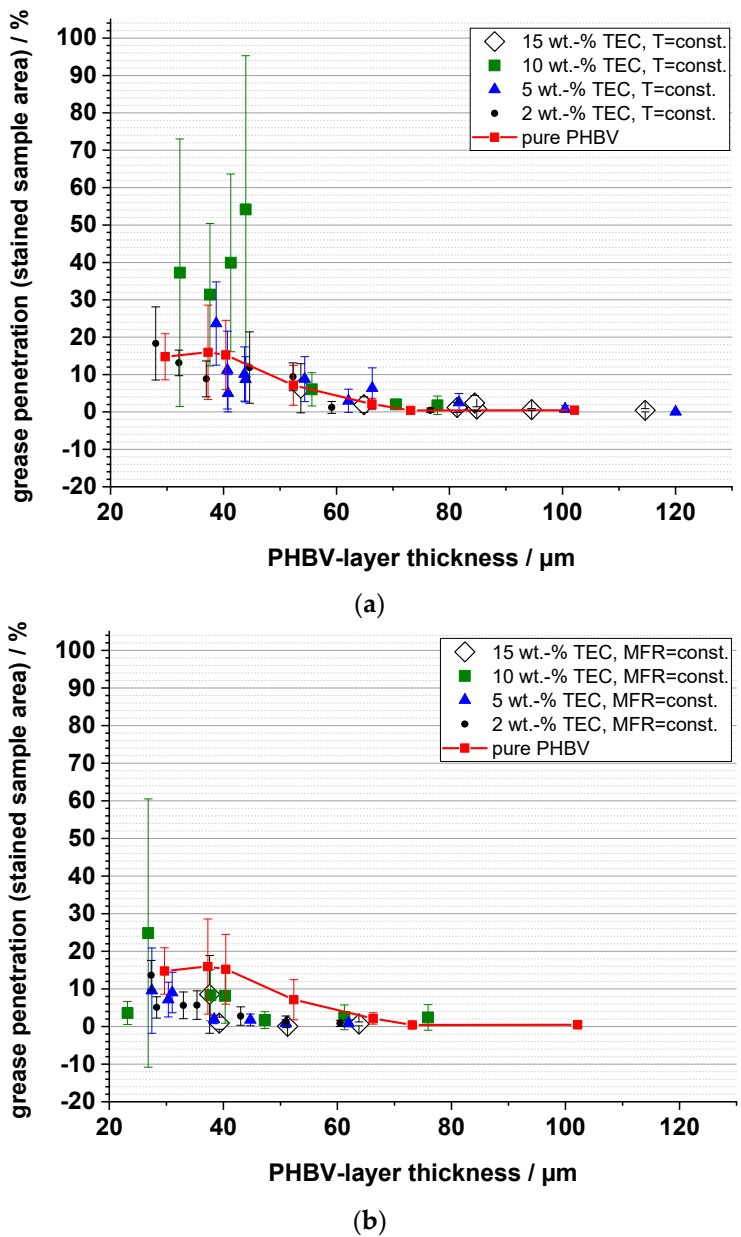

**Figure 14.** Grease barrier (staining) of paper coated with PHBV layers blended with triethyl citrate (TEC): (**a**) temperature (T) during extrusion constant; (**b**) melt flow rate (MFR) during extrusion constant. Thickness: PHBV thickness.

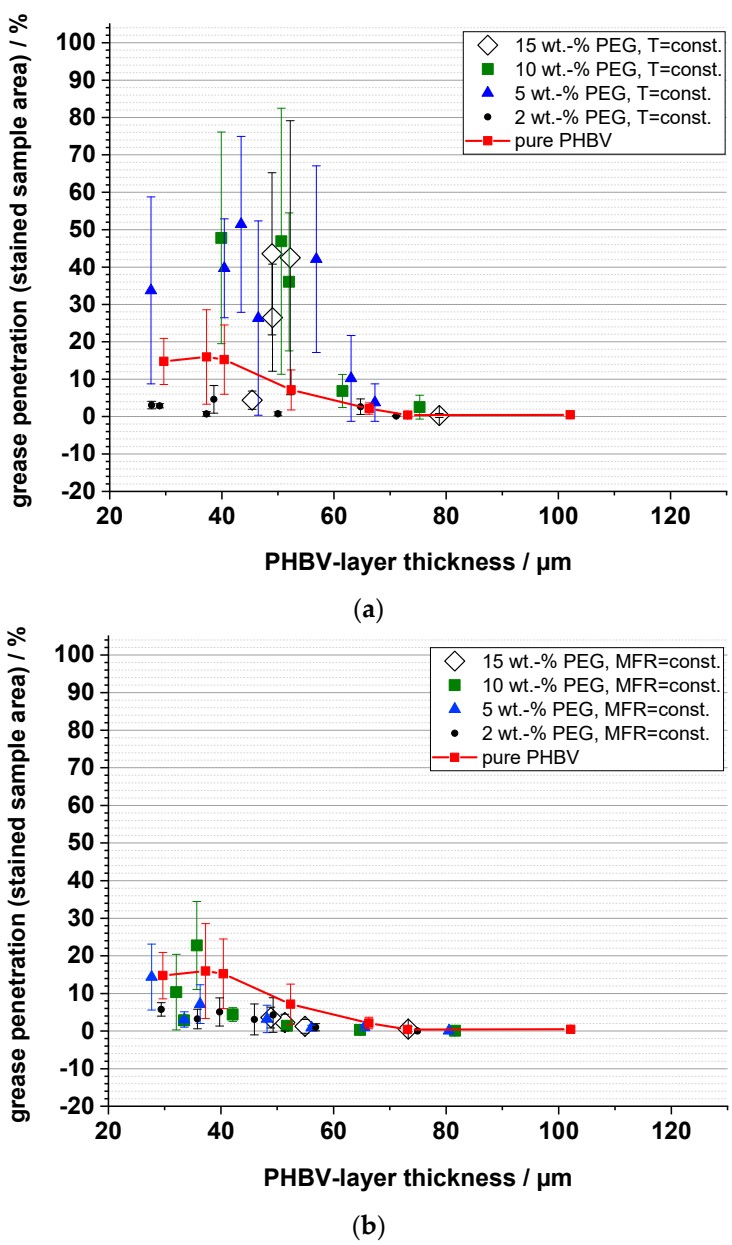

**Figure 15.** Grease barrier (staining) of paper coated with PHBV layers blended with polyethylene glycol (PEG): (**a**) temperature (T) during extrusion constant; (**b**) melt flow rate (MFR) during extrusion constant. Thickness: PHBV thickness.

## 3.7. Surface Roughness

For characterising the surface roughness, *Rz* and *Ra* were analysed. *Rz* is defined as the average distance between the highest and lowest point of each sample, while *Ra* as the arithmetical mean deviation. Blends of PHBV with plasticiser were smoother than pure PHBV film because of the plasticisation and reduction of defects (Figure 16). This is promising when the extrusion coating of thinner PHBV layers is targeted.

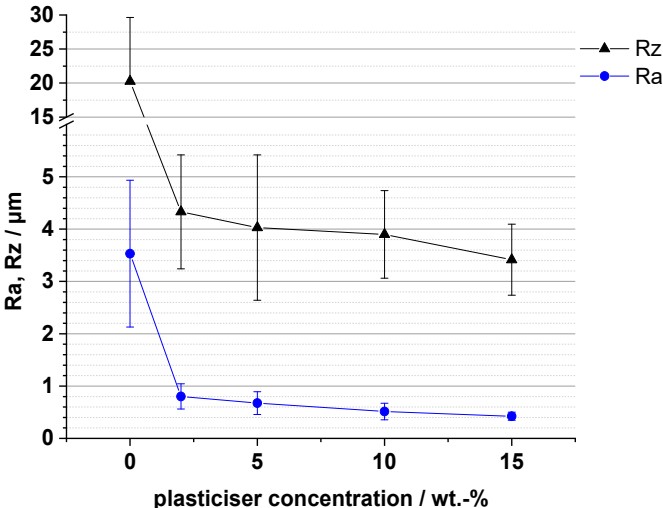

**Figure 16.** Roughnesses of PHBV layers blended with triethyl citrate (TEC), constant MFR (melt flow rate during extrusion).

### 3.8. Microscopic Analysis

The samples were analysed with AFM for the surface structure, with polarisation microscopy for the inside structure and with REM for surface and inside structure. From the AFM measurements (Table 4), an increasing roughness with increasing TEC concentration until 10 wt.% is shown. However, at 15 wt.%, the surface smoothens again.

Interestingly, these result correlate slightly with the results observed with the polarisation microscope (Table 5). However, considering the results of the other samples (Supplementary Material S6), this tendency is not confirmed. Our results are in opposition to the assumption of Rodrigues that with a lower crystallisation temperature, smaller spherulites are visible [22]. The crystallisation temperature reduces with increasing plasticiser concentration (see Section 3.1) compared with Rodrigues [22]. At 500 times magnification, the areas between the crystallites increase.

In the AFM pictures, a further phenomenon was observed. At higher extrusion temperature (constant extrusion temperature) and samples with 15 wt.% plasticiser concentration, irregularities are visible (Figure 17). Sticking out structures are visible. We have no explanation for this phenomenon.

**Table 4.** Atomic force microscopy (AFM) measurements at surfaces of PHBV film blended triethyl citrate (TEC), constant melt flow rate (MFR) during extrusion.

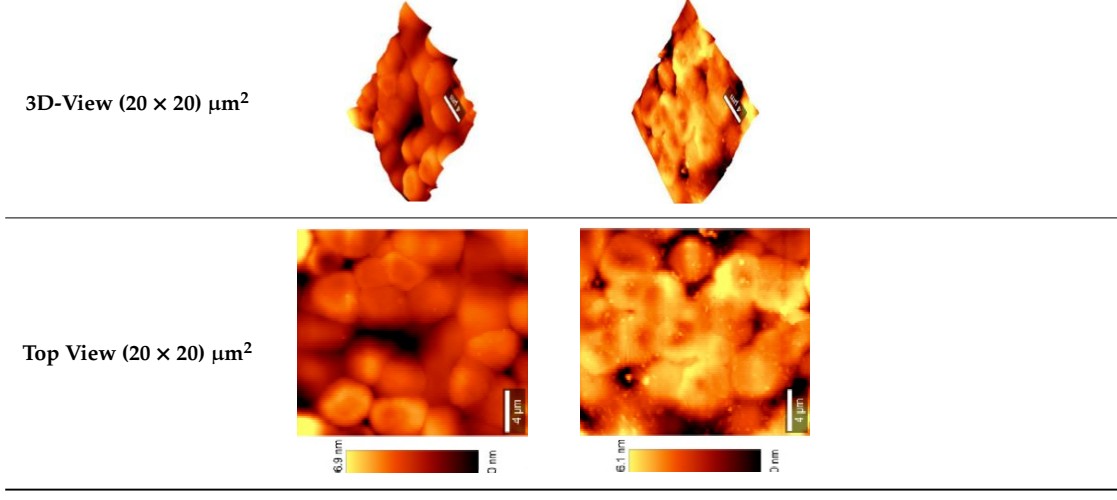

**Table 4.** *Cont.*

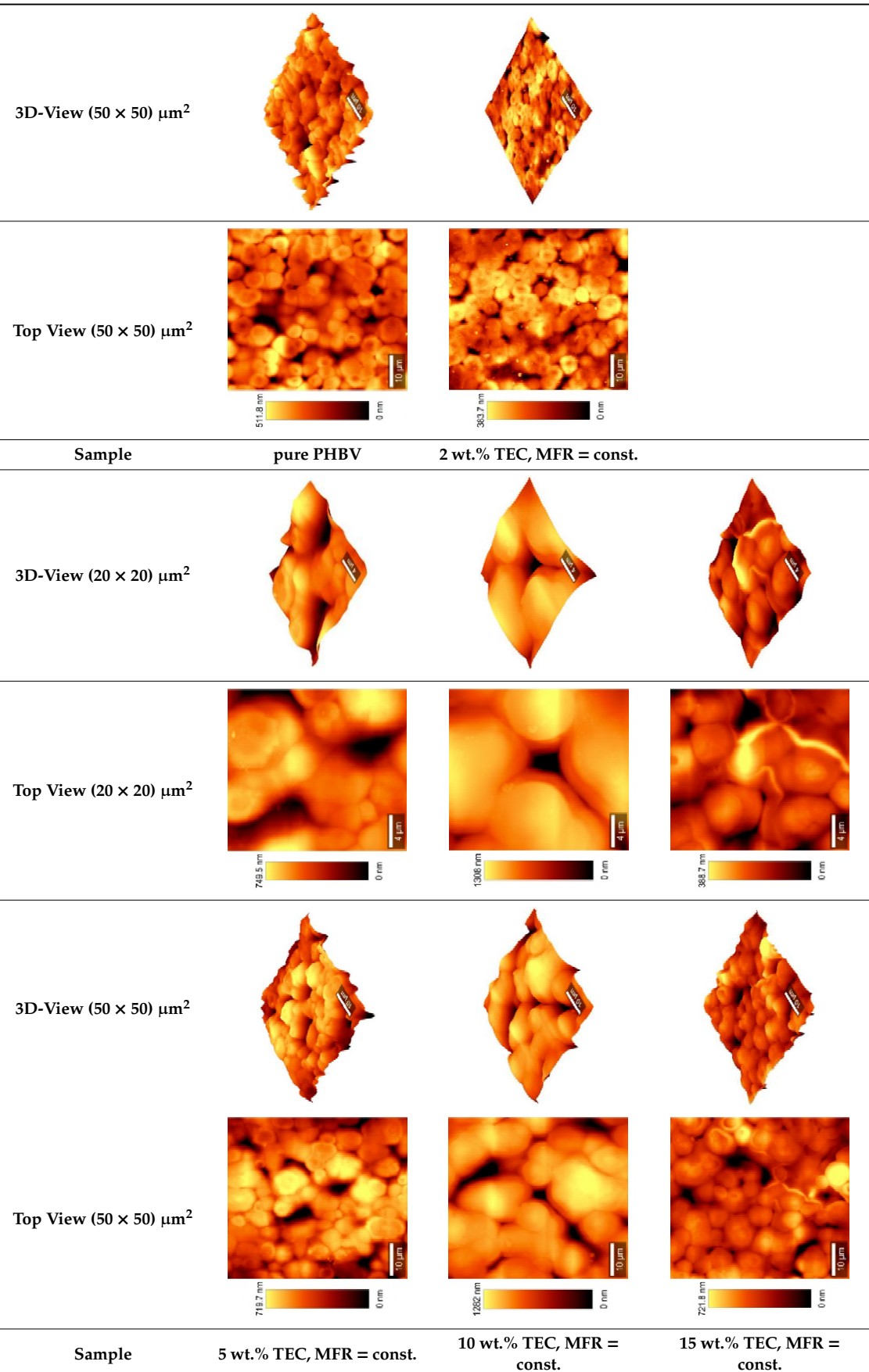

| | pure PHBV | 2 wt.% TEC, MFR = const. | |
|---|---|---|---|
| 3D-View (50 × 50) μm² | | | |
| Top View (50 × 50) μm² | | | |
| **Sample** | **pure PHBV** | **2 wt.% TEC, MFR = const.** | |
| 3D-View (20 × 20) μm² | | | |
| Top View (20 × 20) μm² | | | |
| 3D-View (50 × 50) μm² | | | |
| Top View (50 × 50) μm² | | | |
| **Sample** | **5 wt.% TEC, MFR = const.** | **10 wt.% TEC, MFR = const.** | **15 wt.% TEC, MFR = const.** |

**Table 5.** Polarisation microscopy measurements at surfaces of PHBV film blended triethyl citrate (TEC), constant melt flow rate (MFR) during extrusion. "Anzahl": amount; "Fläche": area.

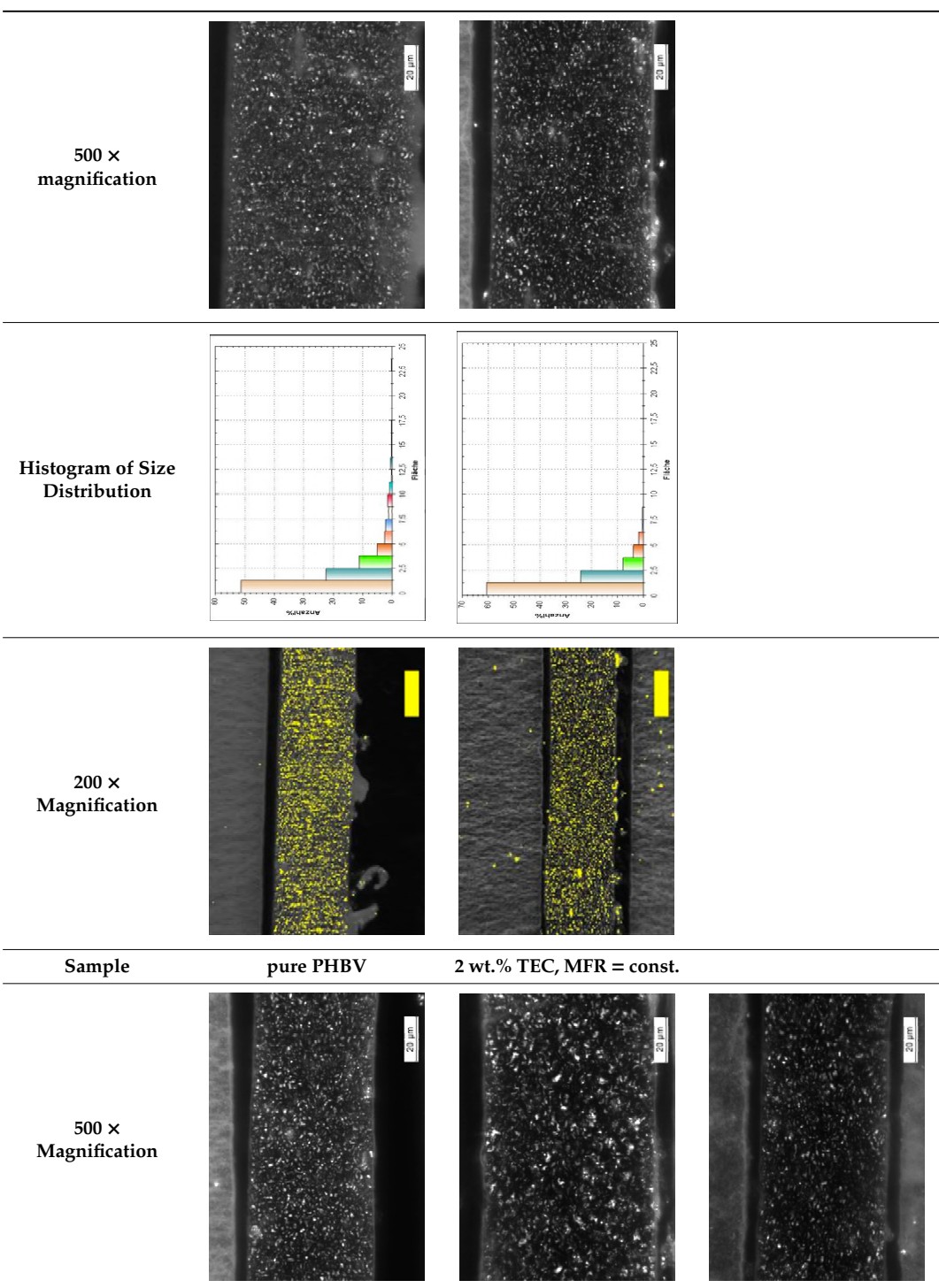

| | pure PHBV | 2 wt.% TEC, MFR = const. | |
|---|---|---|---|
| **500 × magnification** | | | |
| **Histogram of Size Distribution** | | | |
| **200 × Magnification** | | | |
| **Sample** | pure PHBV | 2 wt.% TEC, MFR = const. | |
| **500 × Magnification** | | | |

**Table 5.** *Cont.*

| Histogram of Size Distribution | | | |
|---|---|---|---|
| | 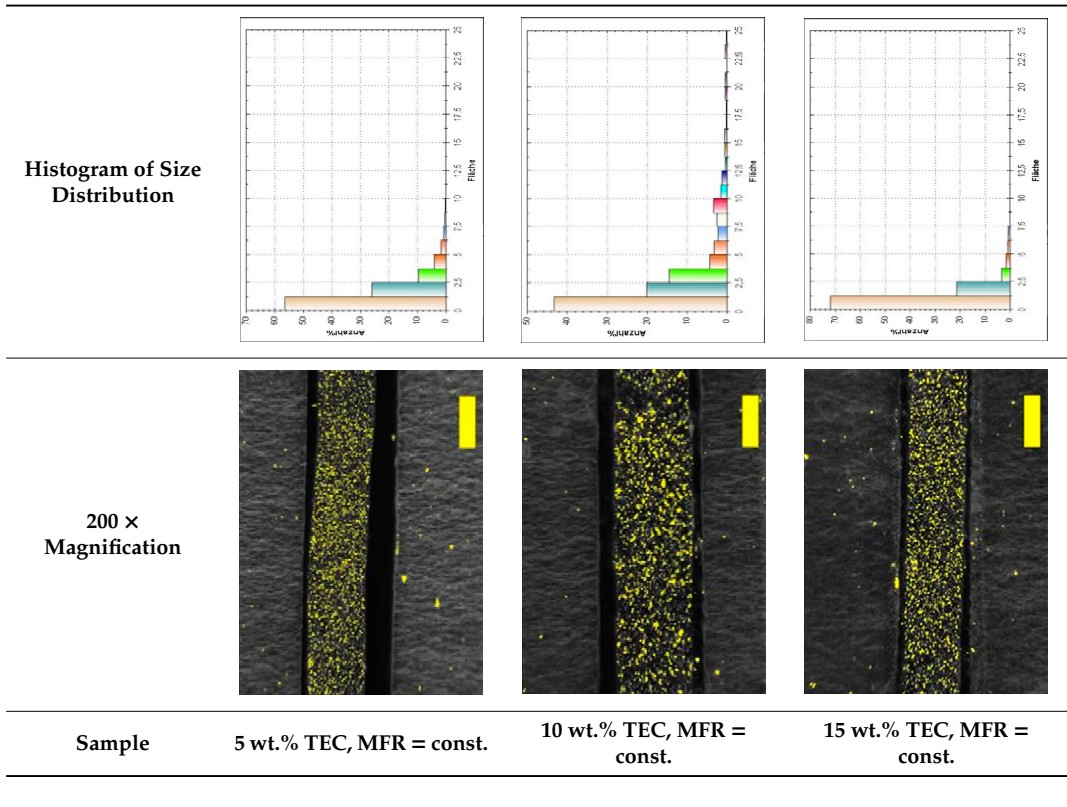 | | |
| **200 × Magnification** | | | |
| **Sample** | **5 wt.% TEC, MFR = const.** | **10 wt.% TEC, MFR = const.** | **15 wt.% TEC, MFR = const.** |

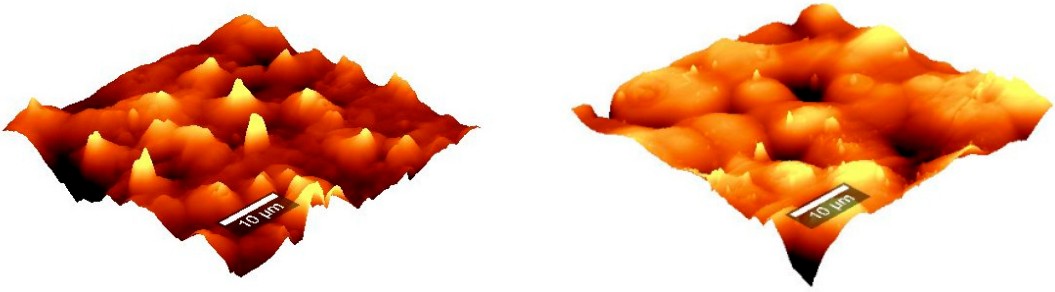

**Figure 17.** Atomic force microscopy (AFM) measurements of PHBV film with 15 wt.% PEG. **left**, extrusion temperature: 176 °C (MFR = const.); **right**, extrusion temperature: 185 °C (T = const.).

SEM analysis was performed on samples with high staining (Table 6). On the surface, defects and cracks are visible at all samples. This can be explained by the low elongation at break and high brittleness of the samples. Defects can be visualized through the entire sample (Figure 18). These defects cause the high staining (low grease barrier). Because a lower film thickness leads to increased staining, we assume defects are more critical for thin films.

**Table 6.** Scanning electron microscopy (SEM) pictures of PHBV films blended with triethyl citrate (TEC).

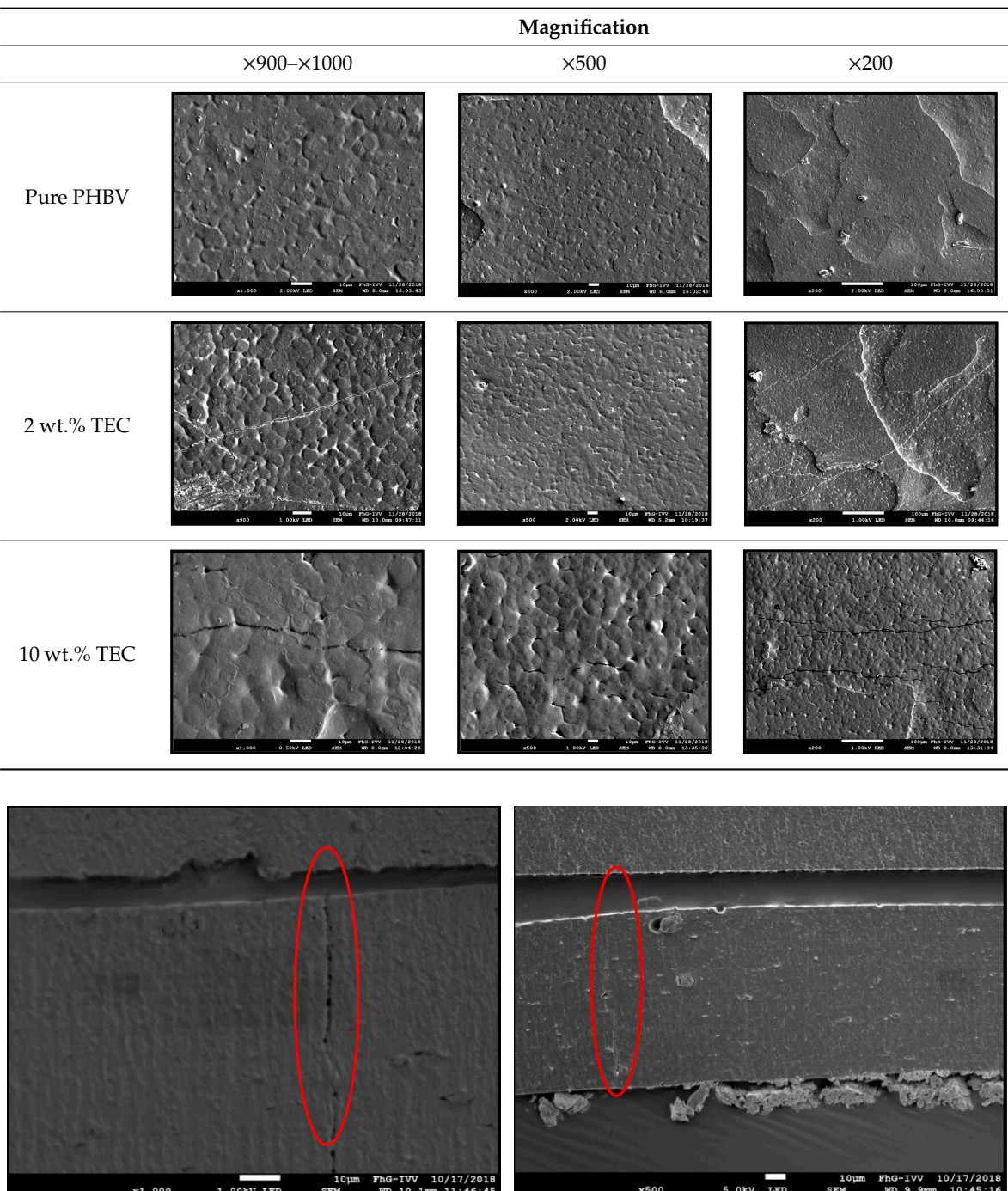

**Figure 18.** Scanning electron microscopy (SEM) pictures of pure PHBV film with defects (marked in red). **Left**: magnification ×1000, **right**: magnification ×500; red ellipse: cracks in the layer.

In the Tables S5–S7 additional results for AFM measurements, polarization microscopic analysis and SEM pictures are presented.

## 4. Conclusions

In this study the effect of TEC and PEG on PHBV processed via extrusion coating on a paper substrate was analysed. By the addition of these plasticisers, the processability and mechanical properties of PHBV based films were improved. By the results is shown that both plasticisers reduce

the crystallinity and increase the flexibility of the films (analysed by a reduced elastic modulus and tensile strength, and increased elongation at break). The bond strength between plasticised PHBV and the paper substrate is sufficient, owing to a cohesion break in the substrate. However, an increase of the plasticiser concentration reduces the bond strength. The increased flexibility of the plasticised PHBV films results in fewer defects and, therefore, an increased grease barrier. The high grease barrier is an important feature for the application as service paper. In order to find a compromise between a suitable flexibility and sufficient bond strength, the addition of a small amount of plasticiser (5 wt.%) is recommended. However, the elongation at break of these samples with approximately 1% is very low and not comparable to conventional polyolefins [40]. The low elongation of break is the reason for cracks and the consequent high grease penetration through these cracks. For packaging related applications, packaging materials should be resistant to grease penetration. Furthermore, we observed a low melt flow stability. Thin PHBV layers of ≤10 μm could not be extruded. For commercial application of PHBV for paper coating, a coating thickness of ≤10 μm would be beneficial to reduce material use and environmental impact.

**Supplementary Materials:** The following are available online at http://www.mdpi.com/2079-6412/9/7/457/s1, Figure S1: RAMAN-spectrum of PHBV, Table S1: Results for the elastic modulus, Table S2: Results for the elongation at break, Table S3: Results for the tensile strength, Table S4: Results of fat penetration (staining), Table S5: Results for AFM measurements, Table S6: Results of the polarisation microscopic analysis, Table S7: SEM pictures.

**Author Contributions:** S.S., M.B. and D.B. designed the experiments. S.S. wrote this manuscript based on the Master Thesis of M.B. M.B., D.B. and N.R. performed the experiments. S.S., M.B., D.B. analysed the data. V.J. consulted us, evaluated our results, and revised and amended the manuscript.

**Funding:** This research was funded by the Federal Ministry of Education and Research (BMBF), German Green Economy lead initiative of the BMBF, Framework Programme "Research for Sustainable Development", funding measure "Plastics in the environment - sources, sinks, solutions" (FONA3), project "Consumer reactions to plastics and their avoidance at the point of sale" ("VerPlaPos"), grant number 01UP1701D.

**Acknowledgments:** The authors thank the Federal Ministry of Education and Research (BMBF) for funding. We thank Daniel Schlemmer, Marius Jesdinszki, Zuzana Scheuerer, and our colleagues for consulting, advice, and support of experiments.

**Conflicts of Interest:** The authors declare no conflict of interest. Some results of this study were already published in the non-peer reviewed magazine Coating International: Sängerlaub, S., Brüggemann, M., Rodler, N., Bauer, D., Extrusion coating of paper with PHBV-an alternative to polyolefins? | [Extrusionsbeschichtung von Papier mit PHBV—Eine Alternative zu Polyolefinen?], Coating International, 52(2), pp. 11–15 [42]. Results presented in this study were used before in the German language master thesis of Marleen Brüggemann with the title "Beschichtung von Papier mit Poly-(Hydroxybutyrate-Co-Hydroxyvalerate) (PHBV)" done at the Fraunhofer Institute for Process Engineering and Packaging IVV, submitted 19.12.2018 at and accepted by the TUM School of Life Sciences Weihenstephan, Chair of Food Packaging Technology, Technical University of Munich.

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
