# Peer review of "Extrusion Coating of Paper with Poly(3-hydroxybutyrate-co-3-hydroxyvalerate) (PHBV)—Packaging Related Functional Properties"

_coatings, doi:10.3390/coatings9070457_

Round 1
Reviewer 1 Report
The article “Extrusion coating of paper with poly(hydroxybutyrate-co-hydroxyvalerate) (PHBV) packaging related functional properties” studies the characteristics of PHBV used as coating polymeric layer on paper. Because of brittleness of this biopolymer, the authors studied the effect of two plasticizers triethyl citrate TEC and polyethylene glycol PEG in formulations at different percentages. A wide characterization has been carried out and also some specific tests of the application as coating have been performed (as grease staining and bond strength) reporting many interesting data. However, the work has some shortcomings and inaccuracies (even the last author “and #” ? seems to be missing) requiring a major revision.
In particular, the abstract has a huge amount of data, sometimes not very significant or unnecessary here; usually only some relevant results are reported in the abstract. Complete data should be reported in the results section.
The description of the test methods should be as more accurate as possible: for example, gaseous nitrogen is in flow and which is the rate? DSC measurement was made with heating and cooling cycles and used the II° heating cycle. Some standard or primary test parameter as temperature of MFR are missing.
Some tests were performed in unusual or unproper way: for example, a high heating rate of TGA (50 °C/min) could hide or shift some thermal phenomena.
Some procedure adopted are not clearly described and could influence the results of the characterization even if statistic technics for the elimination of outliers were adopted (Grubbs test); for example, the peeling could damage the coating or peel off some cellulosic fibers that could work as reinforcement influencing the mechanical tests.
The reference to previous works is undoubtedly an important method of evaluating the results obtained but, in this case, it seems to be excessively emphasized as in lines 222-235 or line366-375.
Captures of the figures should be better organized trying to avoid useless redundancies as for Figures 3,4,5,6 or Figures 22,23,24,25.
Some figures are too small or too large or unreadable in the present format (see Fig. 16,17 Fig. 20,21 or Fig. 18,19,22-25 for each of the wrong formats).
The format of some tables is inadequate for the results reported as Table 2, 4 etc.
About the results there are some discrepancies with the literature that appear singular and should be well checked:
Lines 251-263: how can the author explain that the samples with 2% and 10% of TEC produced at MFR const. show a drop of crystallinity after 28 days and samples with 5% and 15% show only a slight decrease of Xc ?
Line 268 during the II° Heating Tm of neat PHBV decrease probably due to the chain scission degradation caused by the I° cycle; of course, a similar phenomenon affects even the plasticized formulation because PEG does not protect PHBV from degradation.
Figure 7 a double melting peak is expected on PHBV, at least at the second heating because of cooling crystallization and heating crystallization; at least a melting shoulder should be visible...
Table 2 how the author can explain such unexpected results of the onset temperatures of PEG added formulation since in literature there is no effect of PEG300 on degradation. What about PEG 10% test.
Some other improvements could be performed.
I recommend a major revision of the work to make the most of the many useful results obtained.
Author Response
The article “Extrusion coating of paper with poly(hydroxybutyrate-co-hydroxyvalerate) (PHBV) packaging related functional properties” studies the characteristics of PHBV used as coating polymeric layer on paper. Because of brittleness of this biopolymer, the authors studied the effect of two plasticizers triethyl citrate TEC and polyethylene glycol PEG in formulations at different percentages. A wide characterization has been carried out and also some specific tests of the application as coating have been performed (as grease staining and bond strength) reporting many interesting data. However, the work has some shortcomings and inaccuracies (even the last author “and #” ? seems to be missing) requiring a major revision.
For the authors the last comment is not fully clear. # means here that all 3 authors attributed by this sign did the similar amount of work. Sven Sängerlaub und Marleen Brüggemann are first authors, whereas Dieter Bauer is also the supervisor of this work.
In particular, the abstract has a huge amount of data, sometimes not very significant or unnecessary here; usually only some relevant results are reported in the abstract. Complete data should be reported in the results section.
The authors ask the reviewer for his/her approval to keep the amount of specific results in the abstract. The authors are convinced that by providing some more information helps readers to get a good first impression about quantitative results and for ranging the publication about relevance for the reader. For reader it is easier than to decide if the whole publication should be read.
The description of the test methods should be as more accurate as possible: for example, gaseous nitrogen is in flow and which is the rate? DSC measurement was made with heating and cooling cycles and used the II° heating cycle. Some standard or primary test parameter as temperature of MFR are missing.
To use the second cycle at DSC is a standard measure to analyse more standardised samples. An explanatory sentence was added.
The temperature during MFR analysis was varied. An explanatory sentence was added.
More flaws in the experimental description were not found by the authors.
Some tests were performed in unusual or unproper way: for example, a high heating rate of TGA (50 °C/min) could hide or shift some thermal phenomena.
50 °C was used because during extrusion the polymer is heated up quickly for a residence time of several minutes in the extruder. The test was used to get realistically values about thermal stability during extrusion. Worth to mention we wrote in 3.2 that TGA results and extrusion could be directly correlated. Nonetheless, by TGA we yield some data for the evaluation of thermal stability of our used material.
Some procedure adopted are not clearly described and could influence the results of the characterization even if statistic technics for the elimination of outliers were adopted (Grubbs test); for example, the peeling could damage the coating or peel off some cellulosic fibers that could work as reinforcement influencing the mechanical tests.
Grubbs test is a standard test to identify outliers. The main information is that we identified outliers and took them out of the evaluation however in conservative manner, which can be seen by high error bars we present for some results. High error bars are an indication of some influences. We did not elaborate such further to avoid speculation.
Possible fibres during peeling will not reinforce the samples or in a negligible matter because they are on the surface. It is right the taking of samples from the paper could have an influence but we did it in a cautious way. We took samples from the paper the test more realistic samples. Furthermore, we think that by describing the method we provide the reader the possibility to evaluate his/her results with ours also considering possible (not fully quantified) influence of sample preparation.
The reference to previous works is undoubtedly an important method of evaluating the results obtained but, in this case, it seems to be excessively emphasized as in lines 222-235 or line366-375.
The developments on PHBV are ongoing and commercial grads used for such applied studies as in our study can vary. Therefore, we find it useful to refer to previous works often and also highlight differences and similarities from different studies, in the best way as quantitative comparison. Such approach is of value for practical applications.
Captures of the figures should be better organized trying to avoid useless redundancies as for Figures 3,4,5,6 or Figures 22,23,24,25.
The captures were restructured.
Some figures are too small or too large or unreadable in the present format (see Fig. 16,17 Fig. 20,21 or Fig. 18,19,22-25 for each of the wrong formats).
The authors think the journal will care about right formats during layout setting, which is not full control of the authors.
The format of some tables is inadequate for the results reported as Table 2, 4 etc.
The authors think the journal editors will care about right formats during layout setting, which is not full control of the authors.
About the results there are some discrepancies with the literature that appear singular and should be well checked:
Lines 251-263: how can the author explain that the samples with 2% and 10% of TEC produced at MFR const. show a drop of crystallinity after 28 days and samples with 5% and 15% show only a slight decrease of Xc ?
We do not have an explanation for the difference and wrote a sentence about. We think it is worth to present such results to initiate further investigations in following studies.
Line 268 during the II° Heating Tm of neat PHBV decrease probably due to the chain scission degradation caused by the I° cycle; of course, a similar phenomenon affects even the plasticized formulation because PEG does not protect PHBV from degradation.
Chain scission will take place but we were not able to quantify it, e.g. by determination of molecular chain distribution. To avoid speculation we did not discuss the phenomenon further here because reader would than expect some experimental results.
Figure 7 a double melting peak is expected on PHBV, at least at the second heating because of cooling crystallization and heating crystallization; at least a melting shoulder should be visible...
We know such literature data reporting about double peaks. We did not discuss this expected result here further because we would need them to better analyse the molecular structure of the commercial PHBV we used. Of scientific interest is a study about molecular weight distribution and structures of different PHBV with DSC results. Such was not focus of this study.
Table 2 how the author can explain such unexpected results of the onset temperatures of PEG added formulation since in literature there is no effect of PEG300 on degradation. What about PEG 10% test.
We do not have a good explanation for this behaviour. We think such results are interesting for further studies. We assume a time dependant chain scission. A weakness if our study is, as in many studies, that we do not know the molecular weight distribution and status of chain scission of our tested materials. Such determination was not in the intention of this study, which is however of scientific interest. We assume that chain scission before testing is low and comparable at all analysed samples but cannot quantify it.
Some other improvements could be performed.
We did further improvements.
I recommend a major revision of the work to make the most of the many useful results obtained.
Thank you for considering our study as principally worthy for publication.
Reviewer 2 Report
Manuscript ID: Coatings 533152
Extrusion Coating of Paper with Poly(hydroxybutyrate-co-hydroxyvalerate) (PHBV) –Packaging Related Functional Properties
In this manuscript the authors describe the effect of two plasticizers on the properties and, therefore, on the processability of Poly(3-hydroxybutyrate co-3-hydroxivalerate), PHBV, for paper extrusion coating. Material analysis was performed using thermal analysis techniques such as differential scanning calorimetry and thermogravimetric analysis. Several mechanical tests and melt flow measurements were also performed. In addition, measurements of the fat barrier (Staining) and several techniques were used for the morphological characterization of the prepared samples.
The materials prepared and analysed in this paper can be interesting to readers of Coatings, however, several changes should be made before their publication. In general, for the purpose of improving the manuscript quality, several parts of the document should be reformulated.
Introduction section: Line 43-Please put the first time we named it the meaning of the acronym PHBV: ….Poly(3-hydroxybutyrate-co-3-hydroxyvalerate), PHBV, is ……
Material and methods section:
· Page 3, line 90-92: I propose to delete the following sentence: “Jost determined with NMR a value of 11 % [30]. However, it is not clear if both material formulations are identical.) For this reason, a heat of fusion for the calculation of crystallinity of PHB with a value of 146 J/g is taken [31]”, since this information is reported later in the manuscript.
· .Page 3, lines 113-114: Please, explain Why the difference in delivery rates between the two plasticizer?. 30.7, 79.5, 167 and 265.5 ml/h for TEC as plasticizer, front 30, 75, 158.3 and 250 ml/h for PEG.
· Page 4: Please report in Table 1 the values of constant temperature and constant melting flow rate used
· Page 5, line 153: Please, to explain why the second DSC cycle is used in the analysis
· In general, please, to indicate how many samples of each composition were tested in each of the methods of analysis used
Results and Discussion section:
· Page 7, line 222-239: Please rewrite and explain in more detail the effect of the plasticizer on crystallization and melting temperatures. In order to better expose the antecedents I would re-formulate the paragraph from lines 222 to 239. How were these temperatures modified as a function of the time elapsed since the production of the samples? Why were the mean values taken?
· In both figure 1 and 2 replace “…with plasticizers...” by “…with different concentrations of plasticizers…”
· In Figure 2 replace in the ordinate axis Tm by Tc
· Why wasn't the water content determined by DSC?. Were the Tc and Tm values determined on the second DSC scan?
· Page 8, lines 265-267: The following sentence is not clear to me: “The reason why the peak at 40 °C diminished is can be the dissolution and even distribution of PEG in PHBV which could than not crystallize at the quick cooling rate of 10 K/min. “ Can it please be reformulated?
· Page 8, line 264: I suggest change “Samples with 15 wt.% PEG had a peak at circa 40 °C only in the first cycle (Figure 7)” by “Figure 7 shows that the sample with 15wt% PEG have a melting peak close to 40ºC, only in the first cycle”
· Page 8, line 265-267: Reformulate the following sentence: “The reason why the peak at 40 °C diminished is can be the dissolution and even distribution of PEG in PHBV which could than not crystallise at the quick cooling rate of 10 K/min.“ It is not clear what the authors want to indicate with it.
· Page 8, line 268-269: Can you please rephrase or describe the sentence in more detail: “However, in case of a decomposition, the lower Tm of PHBV with 15 wt.% PEG cannot be explained.”
According to my observations, the shape of the PHBV fusion peak changes, it becomes higher and narrower.
· Page 3, Figure 7: Please, put exo or endo address on the thermographs.
· Page 9, line 276-277: Why don't the authors perform TGA under an oxidizing atmosphere (air, or O2)?
· Page 9, Table 2: Perhaps a Figure representation provides information more clearly than a table
· Page 10, line 296-297: Please, provide information about why the increase MFR is stronger with PEG compared to TEC.
· Page 12, line 329: A T constant, by increasing TEC concentration the elongation at break of the films increased, whereas increasing PEG concentration the elongation at break of the films decreased up to 10% plasticizer concentration. Why?. Rephrase the following sentences: “However, at constant extrusion temperature, the PEG-plasticized films had no significant effect in elongation. This might be due to the stronger thermal degradation with this processing. Parra found for blends from PHA and 2 to 10 wt.-% PEG 25 length-% elongation, compared to 9 lengths -% for pure PHA [23].”
· Page 16, Figure 22: Why the error bars are so larger for the sample 10 wt.-% TEC, T=const? The same in Figure 24 for the samples 10 wt.-% TEC, T=const and 5 wt.-% TEC, T=const.
· In my opinion, it would be enriching for students to report an overall conclusion on the effect of the concentration of plasticizers as well as their nature on the morphology of the samples. Thus, a comparison of the results obtained with the different techniques used is necessary.
· Conclusion section: The conclusions should be reformulated after the proposed questions have been clarified.
Minor questions:
· Homogenize the temperature units throughout the manuscript. Always K or always ºC
· Page 11, line 315:Close parentheses after reference 20 “…[20])
· Put reference of Jost paper in the figure caption of figures that report dades of this author.
· Page 15, line 397-398: Why between parentheses? : (The values are the percentage of area that was stained by the coloured peanut oil.)
· Page 22, line 457: The verb is missing in: “Defects can be through the entire sample”. Maybe I should write: “ Defects can be visualized through the entire sample”

Author Response
Extrusion Coating of Paper with Poly(hydroxybutyrate-co-hydroxyvalerate) (PHBV) –Packaging Related Functional Properties
In this manuscript the authors describe the effect of two plasticizers on the properties and, therefore, on the processability of Poly(3-hydroxybutyrate co-3-hydroxivalerate), PHBV, for paper extrusion coating. Material analysis was performed using thermal analysis techniques such as differential scanning calorimetry and thermogravimetric analysis. Several mechanical tests and melt flow measurements were also performed. In addition, measurements of the fat barrier (Staining) and several techniques were used for the morphological characterization of the prepared samples.
The materials prepared and analysed in this paper can be interesting to readers of Coatings, however, several changes should be made before their publication. In general, for the purpose of improving the manuscript quality, several parts of the document should be reformulated
Thank you for advice.
Introduction section: Line 43-Please put the first time we named it the meaning of the acronym PHBV: ….Poly(3-hydroxybutyrate-co-3-hydroxyvalerate), PHBV, is ……
This part adapted now.
Material and methods section:
· Page 3, line 90-92: I propose to delete the following sentence: “Jost determined with NMR a value of 11 % [30]. However, it is not clear if both material formulations are identical.) For this reason, a heat of fusion for the calculation of crystallinity of PHB with a value of 146 J/g is taken [31]”, since this information is reported later in the manuscript.
Here authors propose to keep the sentence. A problem with commercial available polymers is often then formulation can change, also whey the trade name is constant, which is many cases not visible to users. Therefore we find such indication is helpful. The use for the heat of fusion is justified here. Therefore we think it makes sense to mention it here, but also later in the experimental description where an information about is necessary.
· .Page 3, lines 113-114: Please, explain Why the difference in delivery rates between the two plasticizer?. 30.7, 79.5, 167 and 265.5 ml/h for TEC as plasticizer, front 30, 75, 158.3 and 250 ml/h for PEG.
An explanatory sentence was added. The reason is the different density.
· Page 4: Please report in Table 1 the values of constant temperature and constant melting flow rate used
Here we referred to chapter 3.3. and additional information is added.
· Page 5, line 153: Please, to explain why the second DSC cycle is used in the analysis
To use the second cycle at DSC is a standard measure to analyse more standardised samples. An explanatory sentence was added.
· In general, please, to indicate how many samples of each composition were tested in each of the methods of analysis used
Information was added.
Results and Discussion section:
· Page 7, line 222-239: Please rewrite and explain in more detail the effect of the plasticizer on crystallization and melting temperatures. In order to better expose the antecedents I would re-formulate the paragraph from lines 222 to 239. How were these temperatures modified as a function of the time elapsed since the production of the samples? Why were the mean values taken?
We expected a more pronounced time-dependant change of values. But we did not observe much changes and descided not to discuss this point to due to the lack of novelty.
· In both figure 1 and 2 replace “…with plasticizers...” by “…with different concentrations of plasticizers…”
Corrected.
· In Figure 2 replace in the ordinate axis Tm by Tc
Corrected.
· Why wasn't the water content determined by DSC?. Were the Tc and Tm values determined on the second DSC scan?
We did not find our values precise enough for determination of water content. Tc and Tm were determined at 2nd circle. We mentioned it in the methods part.
· Page 8, lines 265-267: The following sentence is not clear to me: “The reason why the peak at 40 °C diminished is can be the dissolution and even distribution of PEG in PHBV which could than not crystallize at the quick cooling rate of 10 K/min. “ Can it please be reformulated?
Reformulated.
· Page 8, line 264: I suggest change “Samples with 15 wt.% PEG had a peak at circa 40 °C only in the first cycle (Figure 7)” by “Figure 7 shows that the sample with 15wt% PEG have a melting peak close to 40ºC, only in the first cycle”
Reformulated.
· Page 8, line 265-267: Reformulate the following sentence: “The reason why the peak at 40 °C diminished is can be the dissolution and even distribution of PEG in PHBV which could than not crystallise at the quick cooling rate of 10 K/min.“ It is not clear what the authors want to indicate with it.
Reformulated.
· Page 8, line 268-269: Can you please rephrase or describe the sentence in more detail: “However, in case of a decomposition, the lower Tm of PHBV with 15 wt.% PEG cannot be explained.”
Reformulated.
According to my observations, the shape of the PHBV fusion peak changes, it becomes higher and narrower.
We overtook it.
· Page 3, Figure 7: Please, put exo or endo address on the thermographs.
Added.
· Page 9, line 276-277: Why don't the authors perform TGA under an oxidizing atmosphere (air, or O2)?
We think during extrusion solved oxygen in PHBV reacts with the polymer and the melt will be free of gaseous oxygen and not be in contact with oxygen.
· Page 9, Table 2: Perhaps a Figure representation provides information more clearly than a table
There is not a clear tendency from these results wherefore find a Table more suitable to avoid over interpretation.
· Page 10, line 296-297: Please, provide information about why the increase MFR is stronger with PEG compared to TEC.
An explanation could be that PEG is less viscous with increasing temperature. However, we did not fully analyse the specific interaction of the used plasticiser with the polymer and did not measure viscosity curves which would be needed for a good discussion. Therefore we want to avoid the proposed discussion without backing of results which we do not have.
· Page 12, line 329: A T constant, by increasing TEC concentration the elongation at break of the films increased, whereas increasing PEG concentration the elongation at break of the films decreased up to 10% plasticizer concentration. Why?.
We do not have a good scientific explanation. Therefore we propose a phenomenological description and avoid too speculative discussion.
Rephrase the following sentences: “However, at constant extrusion temperature, the PEG-plasticized films had no significant effect in elongation. This might be due to the stronger thermal degradation with this processing. Parra found for blends from PHA and 2 to 10 wt.-% PEG 25 length-% elongation, compared to 9 lengths -% for pure PHA [23].”
Reformulated.
· Page 16, Figure 22: Why the error bars are so larger for the sample 10 wt.-% TEC, T=const? The same in Figure 24 for the samples 10 wt.-% TEC, T=const and 5 wt.-% TEC, T=const.
Big error bars are an indication of influences which are not fully reflected. Crease penetration occurs at samples with cracks. I.e. these samples are more inhomogeneous. The exact reason is not clear to us.
· In my opinion, it would be enriching for students to report an overall conclusion on the effect of the concentration of plasticizers as well as their nature on the morphology of the samples. Thus, a comparison of the results obtained with the different techniques used is necessary.
We agree, however we did not do a deep analysis of polymer plasticiser interaction. It was not in focus of this study.
· Conclusion section: The conclusions should be reformulated after the proposed questions have been clarified.
We think the conclusions are sufficient as it is. We want to avoid speculation. The effect of plasticisers is described phenomenological and a deeper discussion is cannot be well backed by our results we have.
Minor questions:
· Homogenize the temperature units throughout the manuscript. Always K or always ºC
We think for DSC “K” is more precise and for temperature °C, because in practical application °C is more of used.
· Page 11, line 315:Close parentheses after reference 20 “…[20])
Done.
· Put reference of Jost paper in the figure caption of figures that report dades of this author.
At many journals it is uncommon because Figures are spread,
· Page 15, line 397-398: Why between parentheses? : (The values are the percentage of area that was stained by the coloured peanut oil.)
Changed.
· Page 22, line 457: The verb is missing in: “Defects can be through the entire sample”. Maybe I should write: “ Defects can be visualized through the entire sample”
Changes.
Round 2
Reviewer 1 Report
The changes made to the revised version of the work “Extrusion Coating of Paper with Poly(3-hydroxybutyrate-co-3-hydroxyvalerate) (PHBV) – Packaging Related Functional Properties” have solved some critical issues reported by the reviewers.
However, some changes indicated were not taken into consideration and remained unresolved. The figures and tables have not been modified and cannot be a part left to the journal, otherwise the reviewer cannot evaluate them.
It would be good to specify that the results that do not have an explanation will be further investigated later to verify their solidity; otherwise it is preferable not to report some data that does not correspond to the literature and to which no scientific explanation can be given.
Avoiding speculation, not discussing the results or not having good explanations for the data obtained in general is not adequate for a scientific publication.
I hope that the authors can remedy these problems.
Figure 11 (previously Figures 18 and 19), such as Figures 14 and 15, are difficult to read in their current form and it would be appropriate to provide these interesting data clearly. At line 277-279 the comment should be clarified: it would seem rather that the melting peak at the second heating becomes larger and narrower because the speed of the thermal ramp has allowed a good crystallization. The commercial grade of the matrix should be specified because the definition of the PHBVs does not correspond to the DSC data or the graph must be removed. A spelling check should be performed.
Hoping that the suggestions could be useful to make the great and important work done by the authors better, a minor but accurate review is required.
Author Response
The changes made to the revised version of the work “Extrusion Coating of Paper with Poly(3-hydroxybutyrate-co-3-hydroxyvalerate) (PHBV) – Packaging Related Functional Properties” have solved some critical issues reported by the reviewers.
Dear Reviewer, thank you for your comments which help us to improve the quality of our manuscript.
However, some changes indicated were not taken into consideration and remained unresolved. The figures and tables have not been modified and cannot be a part left to the journal, otherwise the reviewer cannot evaluate them.
We addressed remarks all of the reviewers. However, in some cases opinions can be different. We would like to mention, that in other publications of MDPI figures are also small. We took these as reference. However, the authors fully agree that figures are not easy to be read. Therefore, we increased their size.
Concerning Table 4 and Table 5 we would like to mention that the final version will be a *.pdf-file where single pages can be easily rotated. Our impression that the formatting as it is now would be acceptable.
It would be good to specify that the results that do not have an explanation will be further investigated later to verify their solidity; otherwise it is preferable not to report some data that does not correspond to the literature and to which no scientific explanation can be given.
We revised the whole manuscript and addressed that right comment.
Avoiding speculation, not discussing the results or not having good explanations for the data obtained in general is not adequate for a scientific publication.
We revised the whole manuscript and addressed that right comment.
I hope that the authors can remedy these problems.
Figure 11 (previously Figures 18 and 19), such as Figures 14 and 15, are difficult to read in their current form and it would be appropriate to provide these interesting data clearly.
We increased the size of these figures to improve readability.
At line 277-279 the comment should be clarified: it would seem rather that the melting peak at the second heating becomes larger and narrower because the speed of the thermal ramp has allowed a good crystallization.
We added this explanation. Thank you.
The commercial grade of the matrix should be specified because the definition of the PHBVs does not correspond to the DSC data or the graph must be removed. A spelling check should be performed.
We specified the used PHBV in section 2.1.1. However, we do not know if and how the producer blended in additives. We also recognized that at some publication DSC analysis of PHBV resulted in double peaks. For us it is an indication of some unknown differences in the material. We think that is worth to study, but we do not have more data for deeper discussion.
Hoping that the suggestions could be useful to make the great and important work done by the authors better, a minor but accurate review is required.
Thank you.
Reviewer 2 Report
Manuscript ID: Coatings 533152R2
Extrusion Coating of Paper with Poly(hydroxybutyrate-co-hydroxyvalerate) (PHBV) –Packaging Related Functional Properties
In this manuscript the authors describe the effect of two plasticizers on the properties and, therefore, on the processability of Poly(3-hydroxybutyrate co-3-hydroxivalerate), PHBV, for paper extrusion coating. Material analysis was performed using thermal analysis techniques such as differential scanning calorimetry and thermogravimetric analysis. Several mechanical tests and melt flow measurements were also performed. In addition, measurements of the fat barrier (Staining) and several techniques were used for the morphological characterization of the prepared samples.
The materials prepared and analysed in this paper can be interesting to readers of Coatings, however, several changes should be made before their publication.
Introduction section:
· Page 1, line 44 : Change “….ppolymer poly(3-hydroxybutyrate-co-3-hydroxyvalerate)…..” by ““….polymer, poly(3-hydroxybutyrate-co-3-hydroxyvalerate)…..”
Material and methods section:
· Page 3, line 91-92: I propose to remove parentheses: “(Jost determined with NMR a value of 11 % [30]. However, it is not clear if both material formulations are identical.)
· Page 3, line 92-93: I do not consider this information: “For this reason, a heat of fusion for the calculation of crystallinity of PHB with a value of 146 J/g is taken [31]”, to be relevant at this point. Leave only the same on line 159 of the manuscript (description of the DSC analysis), since this information is reported later in the manuscript. Put it only once in the manuscript
· Page 3, line 116: To remove parenthesis and put ending point at the end of the sentence.
· Page 4, line 131 : Change subchapter by subsection.
· Page 5, line 154: I insist on always using the same units for the temperature. The same for the rest of the manuscript.
· Page 5, line 154-156: Please, reformulate this paragraph: “triplicates (from day one, 14 and 28 after production), measurements as function of time as single measurements; in two cycles, second cycle used for evaluation.
· Page 5, line 156-158: I propose to remove parentheses: “(The second cycle was used in order to analyse samples exposed to a known thermal treatment and to avoid artefacts such as from thermal parameter variations during processing such as the cooling rate and temperature variations.)”
Results and Discussion section:
• Page 7, line 228-231: if time does not matter, why are samples measured with different times after production?. However, then if you present figure for Tc. Please, to remove these figures (line 266).
· Page 8, line 281: I suggest changing (see next chapter) by (see next subsection)
· Page 9 line 289-291: I suggest removing parenthesis: “…at 270°C, for PHBV with unknown HV concentration. The decomposition temperature decreases by the addition of plasticisers, PHBV with PEG decomposes at 290°C. “
· Page 9, line 292: I suggest changing “PHB-PEG-composite” by “PHB-PEG-blends”.
· Page 9, line 292-293: I suggest removing parenthesis: “However, here the PEG had a molecular weight of 300 g/mol and PHB instead of PHBV was analysed.”
· Page 9, line 294-297: I suggest changing “The HV concentration also effects the decomposition behaviour [39]. The onset temperature of PHBV with a comparable HV content (5%) but from different suppliers varies between 187 and 238°C. An increasing HV content leads to a reduced decomposition temperature [39].” by “The concentration of HV also affects the decomposition behavior, reducing the decomposition temperature as the concentration of HV increases [39]. On the other hand, the starting temperature of the PHBV with a comparable HV content (5%) but from different suppliers varies between 187° and 238°C. The starting temperature of the PHBV with a comparable HV content (5%) but from different suppliers varies between 187° and 238°C.”
· Page 12, line 342-343: I suggest changing “For blends from PHA and 2 to 10 wt.-% PEG elongation was 25 length-%, compared to 9 length-% for pure PHA [23].” by “For blends from PHA and 2 to 10 wt.-% of PEG, elongation was 25 length-%, compared to 9 length- % for pure PHA [23].”
· Page 12, lines 352-354: I suggest changing “Blends with TEC behaved similar to Jost’ samples, but blends PEG dropped whereas the samples of Jost had an increased elastic modulus at higher plasticizer concentrations [27].” by “Blends with TEC behaved similar to Jost’ samples, but whereas for the Jost’ samples an rise of the elastic modulus for higher plasticiser concentrations has been reported, our blends PEG present an opposite behavior [27]”. Please provide some reason for the difference in behavior.
· Page 16, line 426: Change “Figure 26” by “Figure 16”.
· Page 19, line 450: Change “Tabele 5” by “Table 5”.
· Page 21, line 461, 468: Change “Figure 28” by “Figure 18”.
Finally, I suggest that the authors comment on the manuscript of the questions, of my first review, which have not been answered.

Author Response
Extrusion Coating of Paper with Poly(hydroxybutyrate-co-hydroxyvalerate) (PHBV) –Packaging Related Functional Properties
In this manuscript the authors describe the effect of two plasticizers on the properties and, therefore, on the processability of Poly(3-hydroxybutyrate co-3-hydroxivalerate), PHBV, for paper extrusion coating. Material analysis was performed using thermal analysis techniques such as differential scanning calorimetry and thermogravimetric analysis. Several mechanical tests and melt flow measurements were also performed. In addition, measurements of the fat barrier (Staining) and several techniques were used for the morphological characterization of the prepared samples.
The materials prepared and analysed in this paper can be interesting to readers of Coatings, however, several changes should be made before their publication.
Dear reviewer. Thank you for your review which helps us to improve the quality of our manuscript.
Introduction section:
· Page 1, line 44 : Change “….ppolymer poly(3-hydroxybutyrate-co-3-hydroxyvalerate)…..” by ““….polymer, poly(3-hydroxybutyrate-co-3-hydroxyvalerate)…..”
Changed.
Material and methods section:
· Page 3, line 91-92: I propose to remove parentheses: “(Jost determined with NMR a value of 11 % [30]. However, it is not clear if both material formulations are identical.)
Changed.
· Page 3, line 92-93: I do not consider this information: “For this reason, a heat of fusion for the calculation of crystallinity of PHB with a value of 146 J/g is taken [31]”, to be relevant at this point. Leave only the same on line 159 of the manuscript (description of the DSC analysis), since this information is reported later in the manuscript. Put it only once in the manuscript
Changed.
· Page 3, line 116: To remove parenthesis and put ending point at the end of the sentence.
Changed.
· Page 4, line 131 : Change subchapter by subsection.
Changed.
· Page 5, line 154: I insist on always using the same units for the temperature. The same for the rest of the manuscript.
Changed.
· Page 5, line 154-156: Please, reformulate this paragraph: “triplicates (from day one, 14 and 28 after production), measurements as function of time as single measurements; in two cycles, second cycle used for evaluation.
Changed.
· Page 5, line 156-158: I propose to remove parentheses: “(The second cycle was used in order to analyse samples exposed to a known thermal treatment and to avoid artefacts such as from thermal parameter variations during processing such as the cooling rate and temperature variations.)”
Changed.
Results and Discussion section:
• Page 7, line 228-231: if time does not matter, why are samples measured with different times after production?. However, then if you present figure for Tc. Please, to remove these figures (line 266).
We clarified the description.
· Page 8, line 281: I suggest changing (see next chapter) by (see next subsection)
Changed.
· Page 9 line 289-291: I suggest removing parenthesis: “…at 270°C, for PHBV with unknown HV concentration. The decomposition temperature decreases by the addition of plasticisers, PHBV with PEG decomposes at 290°C. “
Changed.
· Page 9, line 292: I suggest changing “PHB-PEG-composite” by “PHB-PEG-blends”.
Changed.
· Page 9, line 292-293: I suggest removing parenthesis: “However, here the PEG had a molecular weight of 300 g/mol and PHB instead of PHBV was analysed.”
Changed.
· Page 9, line 294-297: I suggest changing “The HV concentration also effects the decomposition behaviour [39]. The onset temperature of PHBV with a comparable HV content (5%) but from different suppliers varies between 187 and 238°C. An increasing HV content leads to a reduced decomposition temperature [39].” by “The concentration of HV also affects the decomposition behavior, reducing the decomposition temperature as the concentration of HV increases [39]. On the other hand, the starting temperature of the PHBV with a comparable HV content (5%) but from different suppliers varies between 187° and 238°C. The starting temperature of the PHBV with a comparable HV content (5%) but from different suppliers varies between 187° and 238°C.”
Changed. Thank you for advice.
· Page 12, line 342-343: I suggest changing “For blends from PHA and 2 to 10 wt.-% PEG elongation was 25 length-%, compared to 9 length-% for pure PHA [23].” by “For blends from PHA and 2 to 10 wt.-% of PEG, elongation was 25 length-%, compared to 9 length- % for pure PHA [23].”
Changed. Thank you for advice.
· Page 12, lines 352-354: I suggest changing “Blends with TEC behaved similar to Jost’ samples, but blends PEG dropped whereas the samples of Jost had an increased elastic modulus at higher plasticizer concentrations [27].” by “Blends with TEC behaved similar to Jost’ samples, but whereas for the Jost’ samples an rise of the elastic modulus for higher plasticiser concentrations has been reported, our blends PEG present an opposite behavior [27]”. Please provide some reason for the difference in behavior.
Changed. Thank you for advice.
· Page 16, line 426: Change “Figure 26” by “Figure 16”.
Changed.
· Page 19, line 450: Change “Tabele 5” by “Table 5”.
Changed.
· Page 21, line 461, 468: Change “Figure 28” by “Figure 18”.
Changed.
Finally, I suggest that the authors comment on the manuscript of the questions, of my first review, which have not been answered.
We revised the whole manuscript and addressed that right comment.
Round 3
Reviewer 2 Report
Thank you very much for taking my comments into account. In my opinion, the work can already be published in the current state. However, I urge the authors to review the bibliography. I believe that reference [30] no longer appears in the text of the manuscript and reference [31] should now be renumbered.